# Observing convective activities in complex convective organizations and their contributions to precipitation and anvil cloud amounts

Zhenquan Wang[1, *], Jian Yuan[1]

[1] School of Atmospheric Sciences, Nanjing University, Nanjing, China

*Correspondence to:* Zhenquan Wang (zhqwang@smail.nju.edu.cn)

**Abstract.** The convective processes of precipitation and the production of anvil clouds determine the Earth's water and radiative budgets. However, convection could have very complicated convective organizations and behaviors in the tropics. Many convective activities in various life stages are clustered and connected in complex convective organizations, and distinguishing their behaviors is difficult. In this work, based on hourly infrared brightness temperature (BT) satellite images, with a novel variable-BT tracking algorithm, complex convective organizations are partitioned into organization segments of single cold-core structures as tracking targets. The detailed evolution of the organization structures (e.g., the variation in the cold-core BT, mergers and splits of cold cores) can be tracked, and precipitation and anvil clouds are explicitly associated with unique cold cores. Compared with previous tracking algorithms that focused only on variations in areas, the novel variable-BT tracking algorithm is designed to track the core structure in CCOs and document the evolution of both the area and BT structures. For validation, the tracked motions are compared against the radiosonde cloud-top winds, with a mean speed difference of -1.6 m/s and a mean angle difference of 0.5°.

With the novel variable-BT tracking algorithm, the behaviors of oceanic convection over the tropical western Pacific Ocean are investigated. The results show that the duration, precipitation and anvil amount of lifetime accumulation all have simple loglinear relationships with the cold-core-peak BT. The organization segments of the peak BT values less than 220 K are long-lived, with average durations of 4-16 hours, whereas the organization segments of the warmer-peak BT values disappear rapidly within a few hours but with a high occurrence frequency. The decay process after the cold-core peak contributes to more precipitation and anvil clouds than does the development process. With the core peaking at a colder BT, the differences in the accumulated duration, precipitation and anvil production between the development and decay stages increase exponentially. Additionally, the occurrence frequency of mergers and splits also has a loglinear relationship with the cold-core-peak BT. For the life cycles of the same cold-core-peak BT, the lifetime-accumulated precipitation and anvil amount are strongly enhanced in complicated life cycles with the occurrence of mergers and splits, compared with those with no mergers or splits. For the total tropical convective cloud water budget, long-lived complicated life cycles make the largest contribution to precipitation, whereas long-lived complicated and short-lived simple life cycles make comparable contributions to the anvil cloud amount and are both important.

## 1. Introduction

Precipitation and anvil clouds are two key components of the convective cloud water budget but are usually accompanied by very complicated microphysical and dynamic processes. In climate models, their representations are determined by tunable parameters with large uncertainties, e.g., precipitation and detrainment efficiency (Rennó et al., 1994; Zhao, 2014; Clement and Soden, 2005; Zhao et al., 2016; Suzuki et al., 2013). In cloud-resolving models, the parameterization scheme of convection is still subject to many uncertainties in ice cloud microphysics and subgrid turbulence (Matsui et al., 2009; Blossey et al., 2007; Powell et al., 2012; Bretherton, 2015; Atlas et al., 2024), although cloud dynamics and microphysics can be resolved at fine scales. The challenge is partially because the detailed convection processes of precipitation and the production of anvil clouds have not been sufficiently explored from observations to advance understanding and model parameterization.

The spatial organization of convection varies from a simple isolated cell to a complicated structure that consists of many convective activities in various life stages. The variation in convection organization is closely related to changes in precipitation and the production of anvil clouds (Yuan and Houze, 2010; Yuan et al., 2011; Tobin et al., 2012; Wing and Emanuel, 2014; Mauritsen and Stevens, 2015; Ruppert and Hohenegger, 2018; Bony et al., 2020; Bao and Sherwood, 2019; Houze, 2004). Cloud-resolving models and observations both suggest that convective organizations are important bridges for the interactions between convection and the environment (Tobin et al., 2012; Blossey et al., 2005; Coppin and Bony, 2015; Wing and Emanuel, 2014; Wing et al., 2017; Holloway et al., 2017; Muller and Bony, 2015; Sokol and Hartmann, 2022). Through radiative feedback and circulation, convective organizations are associated with the nonconvecting environment. The drier free troposphere and enhanced radiative cooling of nonconvecting regions can reinforce subsidence to expand the dry region and thereby force the convection in the moist region to aggregate (Blossey et al., 2005; Coppin and Bony, 2015). Over warm oceans, stronger mass convergence and surface turbulent fluxes promote aggregation by developing deep convection and inhibiting scattered convective activities (Coppin and Bony, 2015; Holloway et al., 2017; Wing et al., 2017). Although organizational variations of convection can influence precipitation efficiency (Bao and Sherwood, 2019), when the total atmospheric water amount is not known and is difficult to measure, increased precipitation efficiency might not guarantee a decrease in the anvil cloud amount. Thus, the links among convective organizations, precipitation and anvil clouds still need further observational evidence as constraints for understanding their climate feedback processes.

However, observing the organization and behavior of convection is still challenging. Although active radar and lidar sensors on polar-orbit satellites and ground-based observatories can penetrate convective clouds, their spatiotemporal sampling is too sparse for tracking convection behaviors, without a full picture of the spatial organization of convection. From the images of the brightness temperature at 10.8 μm ($BT_{11}$) of geostationary satellites (GEOs), pixels of thin cirrus clouds cannot be accurately distinguished from cloudless pixels, but the major structure of the organized convection, consisting of the deep convective clouds and their associated anvil clouds, can be observed continuously in time and used for tracking (Richards and Arkin, 1981; Hendon and Woodberry, 1993; Fu et al., 1990). For identifying convection from GEO images, two methods have been proposed in previous studies. One method is to identify the contiguous area under a fixed $BT_{11}$ threshold (Goyens et al., 2011; Schröder et al., 2009; Huang et al., 2018; Williams and Houze, 1987; Chen and Houze, 1997; Kolios and Feidas, 2009; Laing et al., 2008; Feidas and Cartalis, 2007; Fu et al., 2023; Yang et al., 2020; Tsakraklides and Evans, 2003). Based on the fixed threshold, the identified targets usually have complex organizations, but the fixed-threshold method is not capable of being used to further distinguish the detailed convective activities inside complex organizations. In addition, variable-$BT_{11}$ identification has been proposed in recent years, in which a set of adaptive $BT_{11}$ thresholds are used to divide the clustered convection complex into independent convective systems for tracking (Yuan and Houze, 2010; Fiolleau and Roca, 2013; Feng et al., 2023; De Laat et al., 2017; Bouniol et al., 2016; Heikenfeld et al., 2019; Zinner et al., 2008; Zinner et al., 2013). This approach makes it possible to track the detailed variations in complex convection organizations, particularly with respect to tracking the evolution of the $BT_{11}$ structures.

Convective systems can merge and split and their $BT_{11}$ structures can change rapidly. These complicated behaviors make them difficult to track. Most multiple-target tracking algorithms (e.g., multiple-hypothesis tracking) rarely consider mergers and splits and can only be used to track the targets whose number is invariant or merely varies with birth and death events (Blackman, 2004). The aim of storm tracking is to find the associations among convective systems at different times. The most widely used storm-tracking method is based on the overlap in areas between two targets at different times (Williams and Houze, 1987). If convective systems of different times overlap with each other sufficiently, they are associated in time and are deemed to be one storm at different times. This method permits mergers and splits but has flaws in tracking fast-moving storms (Huang et al., 2018). On the other hand, storm positions can be well predicted by matching the $BT_{11}$ patterns in the latter image via cross correlation (Leese et al., 1971; Nieman et al., 1997; Velden et al., 1998; Salonen and Bormann, 2016; Hersbach et al., 2020). This method has been widely applied to in atmospheric motion vector observations (Salonen and Bormann, 2016),

which are among the most important data sources for assimilation into reanalysis winds (Hersbach et al., 2020). However, the association of convective systems at different times is difficult to determine by their positions when they merge and split. Nevertheless, these two methods are complementary and can be combined to first derive storm displacements and then determine the temporal associations between convective systems according to the dynamic overlap (i.e., the overlap in areas after moving it to the position predicted by cross correlation) (Feng et al., 2023; Zinner et al., 2013).

In this work, complex convective organizations (CCOs) are segmented into simple structural components of single cold cores and tracked separately on the basis of a novel variable-$BT_{11}$ tracking algorithm. Compared with fixed-threshold tracking, variable-$BT_{11}$ tracking has the advantages of documenting more detailed convective evolution in CCOs. Although several variable-$BT_{11}$ tracking algorithms have been proposed, the tracked life cycle is still described mostly by the variation in areas and lacks $BT_{11}$ structural information. *This novel variable-$BT_{11}$ tracking algorithm developed in this work has the capability of tracking the $BT_{11}$ structural evolution in CCOs.* The precipitation and non-precipitating anvil clouds in CCOs are explicitly associated with unique cold cores with a well-organized structure in the novel tracking algorithm.

This paper is laid out as follows: Sect. 2 describes the data and methods used in our analyses. Sect. 3 introduces the novel variable-$BT_{11}$ segment tracking algorithm and its comparison to fixed-threshold tracking. Sect. 4 explores the relationships of the convection duration, precipitation and anvil production with the $BT_{11}$ structures. Sect. 5 presents conclusions.

## 2. Data and methods

### 2.1 Images from GEOs

The $BT_{11}$ images in the tropics between 20°S-20°N and 90°W-170°E were taken by radiometer imagers on geostationary Multi-functional Transport Satellite 1 Replacement and 2 Replacement (MTSAT-1R and -2R), with a scanning start time of half an hour and view zenith angles of less than 60°. The data of those $BT_{11}$ images from 2006 with 1-hour and 8-km resolutions were obtained from the Satellite ClOud and Radiative Property retrieval System (SatCORPS) of the Clouds and the Earth's Radiant Energy System (CERES) project. In the CERES SatCORPS, the $BT_{11}$ was calibrated against the Moderate Resolution Imaging Spectroradiometer (MODIS) from the Aqua (Doelling et al., 2013; Doelling et al., 2016). To facilitate data processing, the $BT_{11}$ images of 8-km pixels were further gridded to 0.05° via linear interpolation (Amidror, 2002).

### 2.2 Global precipitation measurement (GPM)

At fine scales (0.1° and half-hour resolution), the GPM data combine all available sensors for precipitation estimates, which include microwave imagers from multiple low-Earth orbit satellites, the infrared (IR) channel of GEO radiometers and land-surface rain gauges (Huffman et al., 2007; Huffman et al., 1997). The microwave brightness temperature is sensitive to atmospheric hydrometers of precipitation but has sparse spatiotemporal sampling due to its sun-synchronous orbit. For those grids without microwave observations, the GEO-IR $BT_{11}$ was used to estimate precipitation according to the spatially varying calibration coefficient of the microwave precipitation rates (Huffman et al., 1997). To improve accuracy, rain gauges were further used to rescale satellite estimates of precipitation rates (Huffman et al., 1997). It has been demonstrated that this satellite-based precipitation product performs well for strong precipitation events with a mean bias smaller than 1 mm/day but misses 20-80% of the light precipitation (< 10 mm/day) (Tian et al., 2009). In the tropics, light precipitation (< 1 mm/hour) accounts for approximately 55-70% of the precipitation area but contributes to only 9-18% of the total precipitation (Yuan and Houze, 2010). Only precipitation rates greater than 1 mm/hour are considered the regions with precipitation in this work since light precipitation has high uncertainty in the GPM data and relatively low contributions to total precipitation.

### 2.3 Cloud-top winds from ground-based radar and radiosonde observations

Cloud detection from radar and wind observations from radiosondes were combined to derive cloud-top winds at three tropical ground-based observatories of the Atmospheric Radiation Measurement (ARM) program: Darwin (130.9°E, 12.4°S), Manus Island (147.4°E, 2.1°S) and Nauru Island (166.9°E, 0.5°S). The vertical distribution of hydrometers up to 20 km above the ground was detected via 35 GHz millimeter-wave cloud radar (MMCR), with temporal and spatial resolutions of 10 s and

45 m, respectively. The best estimate reflectivity of the MMCR in the range of -50 to 20 dBZ was provided in the ARM program Active Remote Sensing of Clouds (ARSCL) value-added product at the three sites. A reflectivity higher than -40 dBZ was identified as a cloud (Zhao et al., 2017). To match the GEO observations, instantaneous cloud profiles within 5 minutes around the time of the GEO radiometer imager scanning at these three sites were collected to compute the cloud fraction. The 10-min cloud-fraction profile was computed as the ratio of the number of cloud occurrences to the total number of observations at each height. Continuous levels of cloud fraction greater than zero were identified as cloud layers. The thickest high cloud layer with a top greater than 5 km and a maximum cloud fraction of at least 50% was selected as the major high cloud layer passing over the sites. The cloud top refers to the uppermost height of the major high cloud layer.

Winds were detected by the ARM balloon-borne radiosondes with high vertical and temporal resolutions of 10 m and 2.5 s, respectively. The accuracy of the radiosonde wind speed was approximately 0.5 m/s. The radiosondes were launched two times a day at Manus and Nauru (approximately 11:30 and 23:30 UTC) and four times a day at Darwin (approximately 4:30, 11:15, 16:30 and 23:15 UTC). Notably, balloon-borne radiosondes usually take hours and drift dozens of kilometers away from the launch location to approach the cloud top height. To derive cloud-top winds, the time difference between the balloon-borne radiosondes reaching the cloud top height and the MMCR observations of the cloud top must be within one hour for quality control.

### 2.4 Comparison of cloud-top winds and tracked cloud motions

The difference between the observed cloud-top winds and the tracked cloud motions is assessed as follows, which is consistent with those in Nieman et al. (1997):

$$Speed\ bias = \frac{1}{N}\sum_{i=1}^{N}\left(\sqrt{U_i^2 + V_i^2} - \sqrt{U_r^2 + V_r^2}\right), \tag{1}$$

$$Angle\ bias = \frac{1}{N}\sum_{i=1}^{N}\left(\arctan\left(\frac{V_i}{U_i}\right) - \arctan\left(\frac{V_r}{U_r}\right)\right), \tag{2}$$

$$MVD = \frac{1}{N}\sum_{i=1}^{N}\sqrt{(U_i - U_r)^2 + (V_i - V_r)^2}, \tag{3}$$

$$SD = \sqrt{\frac{1}{N}\sum_{i=1}^{N}(\sqrt{(U_i - U_r)^2 + (V_i - V_r)^2} - MVD)^2}, \tag{4}$$

$$RMSE = \sqrt{MVD^2 + SD^2}. \tag{5}$$

Here, the mean speed and angle bias, the mean vector difference (MVD), the standard deviation (SD) of the MVD and the root-mean-square error (RMSE) of the tracked cloud motions compared with the observational cloud-top winds were computed. U and V are the x- and y-component winds, respectively. The subscripts i and r indicate an individual sample of the tracked cloud motion and the corresponding reference cloud-top winds of radiosondes, respectively, and N is the total number of samples.

### 2.5 $t$ test and confidence intervals

The 95% confidence interval for the mean value was computed via the t test: $\bar{x} \pm t_c \frac{s}{\sqrt{N}}$, where $\bar{x}$ is the mean value of all samples; $t_c$ is the critical value for t; and s is the standard deviation of all the samples. N is the number of independent samples, and N is determined by the sample length divided by the distance between independent samples (Bretherton et al., 1999).

## 3. Tracking the convective organization segments

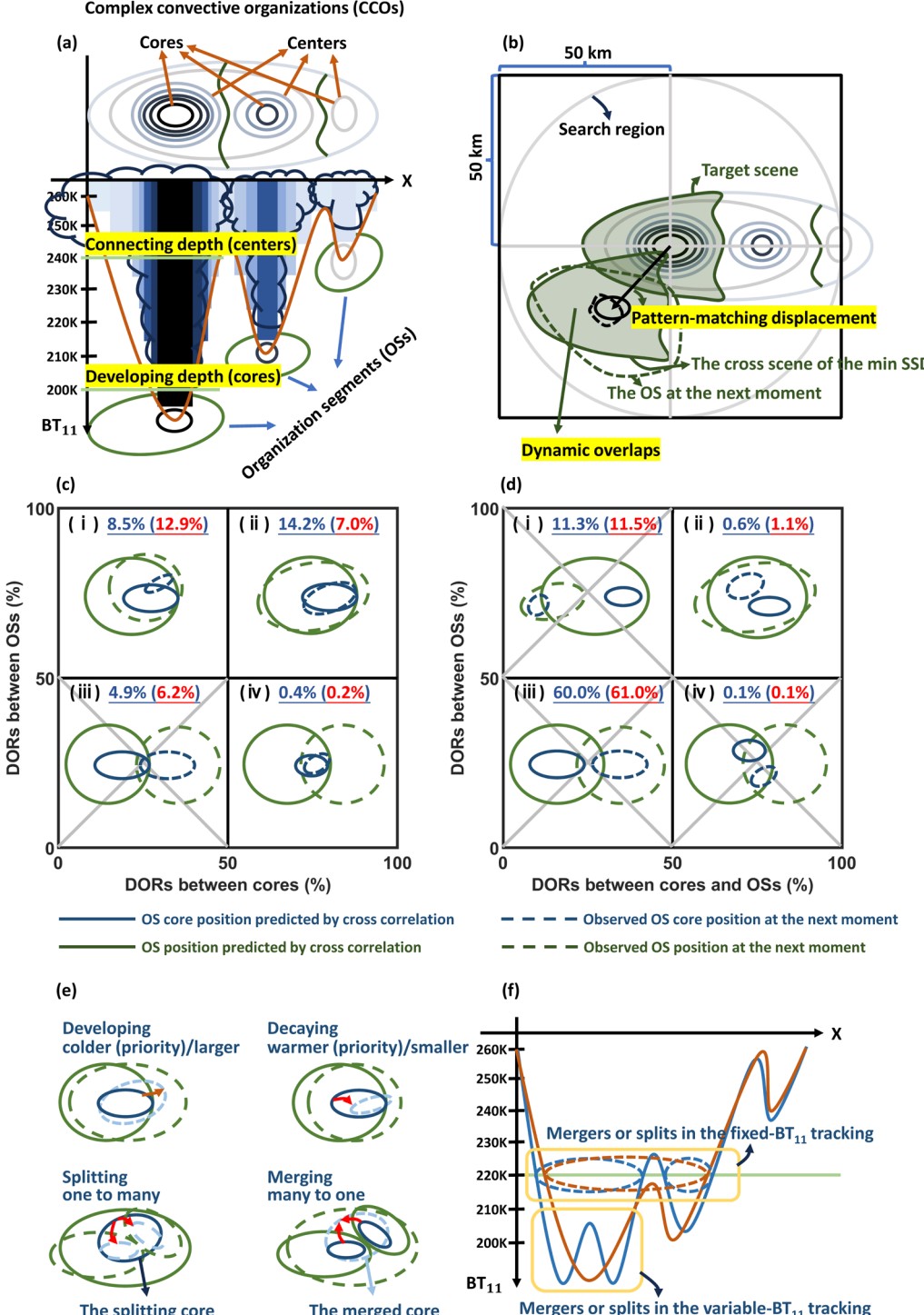

**Figure 1.** Schematic illustrations of the variable-BT$_{11}$ segment tracking algorithm. (a) Example illustrations of segmenting the CCO into single-core OSs as tracking targets. The CCO 3-dimensional structures in x, y and BT$_{11}$ are identified by the adaptive variable-BT$_{11}$ thresholds. The cold-core BT$_{11}$ indicates the depth of development. The cold-center BT$_{11}$ indicates the depth of the connection. (b) Example illustrations of tracking the OS by combining cross correlation and overlap in areas. The OS is moved to the position predicted by cross correlation and then overlaps with the OSs at the next moment. (c-d) Dynamic overlapping situations of two OSs of different moments when their cores have overlaps and no overlaps, respectively. The solid blue and green lines indicate the core and OS of the current moment at the position predicted by cross correlation, respectively. The dashed blue and green lines indicate the core and OS positions of the next moment, respectively. The gray cross indicates none of the association between OSs. The occurrence frequency of each condition is listed at the top of each subpanel by blue numbers. The frequency for overlaps without consideration of the OS movement is listed in parentheses by red numbers. (e) Examples illustrating the tracked structural evolution of OSs (i.e., development, decay, mergers and splits). The red arrows indicate the evolution of the OS with time. (f) Illustrations of the difference between the variable-BT$_{11}$ and fixed-BT$_{11}$ tracking for merger and split events. The solid red and blue lines are the CCO BT$_{11}$ structures at different times captured by the adaptive variable-BT$_{11}$ thresholds. The dashed red and blue contours represent the mergers and splits in the fixed-threshold tracking of 220 K.

To distinguish clustered convective activities in CCOs, the organization segments (OSs) of single but variable-$BT_{11}$ cold cores are partitioned from CCOs (Fig. 1a) and are tracked by combining the cross correlation and the area overlap (Fig. 1b) on the basis of the hourly infrared satellite images. This novel variable-$BT_{11}$ segment tracking algorithm and its difference from the conventional fixed-threshold tracking algorithm are introduced in this section as follows.

### 3.1  Segmenting CCOs into the OSs of single cold cores

As illustrated in Fig. 1a, the CCO is the complex organization of multiple connected convections and is identified by the contiguous area of the $BT_{11}$ colder than 260 K. The 260-K threshold of $BT_{11}$ is normally used to identify the pixels with high clouds (Minnis et al., 2008; Minnis et al., 2011). The 260-K isotherm can enclose 95% of deep convective clouds and as much of the anvil cloud as possible but with the least contamination from lower-level clouds (Yuan and Houze, 2010; Yuan et al., 2011; Chen and Houze, 1997). To identify the organized structures of CCOs, a set of adaptively variable-$BT_{11}$ thresholds from

180-260 K per 5-K interval and a minimum area threshold of 1000 km$^2$ are used to capture the "growth rings" in CCOs. As shown in Fig. 1a, these rings reflect the CCO structure in three dimensions, namely x, y and $BT_{11}$, and are the fundamental indicators of its internal dynamics (Houze, 2004). The innermost ring of the coldest local $BT_{11}$ is defined as the cold core, which is the most active vertically developing region. The $BT_{11}$ of the cold core represents the depth at which the convection developed. For a CCO consisting of multiple cold cores, the $BT_{11}$ of the coldest core in the CCO is defined as the CCO $BT_{11}$

for representing the depth of CCO development. The isolated ring of the warmest $BT_{11}$ that encloses only one cold core is the cold center. The convection would be connected (disconnected) to the surrounding convections outside (inside) the center. Thus, the cold-center $BT_{11}$ can be used to indicate the connecting condition between the convections in the CCO.

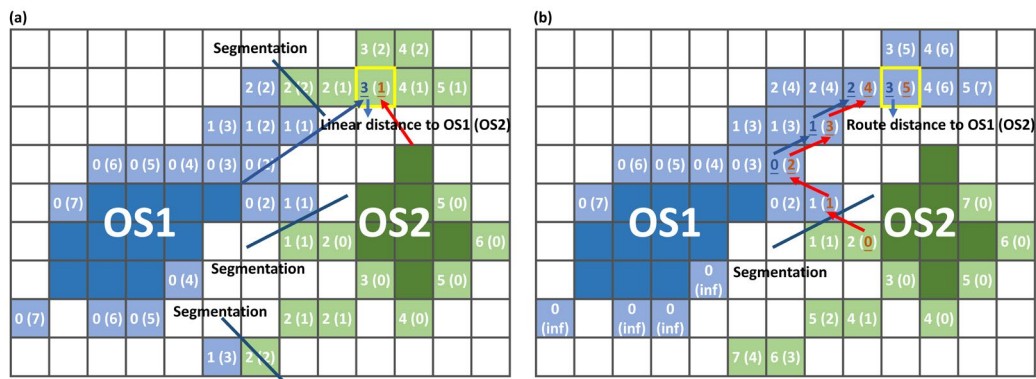

Figure 2. Illustrations of segmentation according to the nearest linear distance (a) and the nearest route distance (b). The dark
blue and green pixels represent the OS1 and OS2 centers, respectively. The colored pixels outside the centers are the pixels to be assigned in the contour of the cold-center $BT_{11}$ plus 1 K. The light blue and green pixels are assigned to OS1 and OS2, respectively. The numbers inside those pixels indicate the number of necessary pixels to connect with OS1 and OS2, respectively. The arrows in (a) and (b) represent the nearest distances of OS1 and OS2 to reach the yellow-edge pixel, as examples to illustrate the computations of the linear distance and the route distance, respectively.


    The segmented single-core structural component of CCOs is defined as the OS, as illustrated in Fig. 1a. To partition the CCO into OSs, the pixels lying outside the centers are assigned to the connected neighborhood OSs by the 1-K interval. To be specific, all $BT_{11}$ contours of the 1-K interval between the cold-center $BT_{11}$ and 260 K need to be found first. The assignment of the pixels outside the centers is conducted in the order from cold to warm $BT_{11}$ contours of the 1-K interval. The initial OS

is just the center and it is updated after every 1-K-interval assignment. An example illustration of the 1-K-interval assignment is shown in Fig. 2. On the basis of the 8-point-connected neighborhood in which the 8 surrounding points are recognized as the neighborhood connected to the center point, the distance between two pixels is computed as the number of necessary pixels connecting them. According to the nearest linear distance, as shown in Fig. 2a, some of the pixels assigned to OS2 (those light green pixels in Fig. 2a) are disconnected from OS2 but connected to OS1. After the assignment, OS2 is composed of two

disconnected parts. For an organized convective system, the assigned pixels outside the center can also be understood as outflowing anvil clouds from the center. It would be strange that the outflowing anvil clouds from OS2 are not connected with

its origin but connected with OS1. To avoid these conditions, the distance of the nearest route is used to determine the pixel assignment. Here, the route of OS1 and OS2 to reach a pixel (the blue and red arrows in Fig. 2b) is confined to be within the 1-K-interval contour. Pixels of the same distance to OS1 and OS2 are randomly assigned. In Fig. 2b, the assignment of the pixels on the basis of the distance of the nearest route is more reasonable than that in Fig. 2a on the basis of the nearest linear distance. Thus, in every 1-K-interval assignment, the distance of the nearest route is used to accomplish the segmentation and the OSs are updated with these newly assigned pixels iteratively until all the pixels within the CCO are assigned.

The final OS is the 3-dimensional (x, y and $BT_{11}$) structure of a single cold core. In the dimension of $BT_{11}$, the cold-core $BT_{11}$ can represent its development depth and the cold-center $BT_{11}$ can indicate its connecting conditions with surrounding OSs in CCOs. In the horizontal dimensions of x and y, the OS area can be further separated into precipitating and non-precipitating (precipitation less than 1 mm/hour) regions on the basis of the GPM. The non-precipitating area is identified as the anvil cloud. With segmentation, those precipitation and anvil pixels are explicitly associated with unique cold cores. These key definitions of variable-$BT_{11}$ tracking are depicted in Fig. 1a and summarized in Table 1 for easy checking.

**Table 1.** Summary of the key definitions for variable-$BT_{11}$ tracking developed in this study

| Name | Definition |
|---|---|
| Complex convective organizations (CCOs) | The contiguous area of the $BT_{11}$ colder than 260 K. |
| Organization segments (OSs) | The segmented single-core structural component of CCOs. |
| Cold-core $BT_{11}$ (OS developing depth) | The local coldest $BT_{11}$ contour in OSs. |
| Cold-center $BT_{11}$ (OS connecting depth) | The local warmest isolated $BT_{11}$ contour of only enclosing one core in OSs. |
| CCO $BT_{11}$ (CCO developing depth) | The coldest cold-core $BT_{11}$ of multiple cores in the CCO. |
| Anvil clouds | The non-precipitating (precipitation less than 1 mm/hour) region of each OS. |
| Dynamic overlapping rates (DORs) | The overlapping rates in areas after moving it to the position predicted by cross correlation. |
| Merger and split $BT_{11}$ | The $BT_{11}$ of the merged cold core and the $BT_{11}$ of the splitting cold core. |
| Cold-core-peak $BT_{11}$ | The coldest cold-core $BT_{11}$ in life cycles, representing the convective peaking strength. |
| Development and decay stages | The stage before and after the time of the cold core peaking at the coldest $BT_{11}$ (if there are multiple cores of the same coldest $BT_{11}$ in the life cycle, the one of the largest core areas is selected). |
| Lifetime-accumulated precipitation and anvil cloud amount | The sum of the observed OS precipitation and anvil in hourly satellite images during its lifetime. |

### 3.2 Tracking the displacement of OSs on the basis of cross correlation

The OS displacement is derived by searching for the maximum similarity of its $BT_{11}$ pattern in the next GEO-IR image via cross correlation (Leese et al., 1971; Velden et al., 1998). As shown in Fig. 1b, the target scene is the OS $BT_{11}$ pattern. The search region is centered at the core centroid of the target and confined to a radius of 50 km, which corresponds to a maximum cloud-drift motion of 50 km/hour (Merrill et al., 1991). The cross scene has the same shape as the OS target and refers to all possible scenes to match the OS target within the search region. The $BT_{11}$ pattern of the target scene is normalized, as is the $BT_{11}$ pattern of each cross scene. The patten-matching displacement is determined by the minimum of the sum of squared differences (SSD) of the normalized $BT_{11}$ between the OS target scene and the cross scene:

$$SSD = \sum_{x,y}[BT'_{11}(x,y) - \widetilde{BT}'_{11}(x,y)]^2, \tag{6}$$

where $BT'_{11}(x,y)$ and $\widetilde{BT}'_{11}(x,y)$ are the normalized $BT_{11}$ values at pixel (x, y) of the target scene and the cross scene in the

search, respectively. Here, the minimum SSD corresponds to the maximum pattern correlation. The final match is examined by the pattern correlation coefficient. For the OS of areas larger (smaller) than 5000 $km^2$, the match is valid when the correlation of the pattern is greater than 0.6 (0.8). The correlation threshold values are consistent with those in Daniels et al. (2020).

Otherwise, the OS $BT_{11}$ structures would change rapidly in one hour and would rather be considered stationary.

### 3.3 Tracking OSs via dynamic overlaps

In Fig. 1b, to track the temporal evolution of OSs, the OS is moved to the location predicted by cross correlation and then overlaps with the OSs in the next satellite image. In this way, dynamic overlaps can be used to tolerate the fast-moving OS in tracking. For the OS with the core structure, three indices of the dynamic overlapping ratio (DOR) are considered to determine

the associations of two OSs at different times, including the DOR between cores, the DOR between OSs and the DOR between cores and OSs. The DOR between cores is the ratio of their overlaps in cores relative to the minimum area of the cores to represent the degree of core overlap. The DOR between OSs is the ratio of the OS overlap relative to the minimum area of the OS to represent the degree of OS overlap. The DOR between cores and OSs is the ratio of the overlap of the OS to the core relative to the core area, which represents the degree of the core overlapped by the OS.

Two OSs of different moments are associated in time and considered the same one OS at different times when these two OSs overlap sufficiently. With the dynamic overlap, an OS is moved to the new predicted location via cross correlation to overlap with the OSs at the next moment. In this case, a necessary condition to consider the associations of the OSs at the next moment to the OS is that their DORs are at least greater than zero. After moving to the new predicted location, an OS might overlap with many OSs of the next moment simultaneously, and the overlapping situations are various. For instance, one OS

can have large overlaps with the major core structure of an OS of the next moment, and meanwhile it can also have some overlaps with the margins of another OS of the next moment. These three DOR indices can be used to identify these distinct overlapping conditions from the overlapping degrees of their cores and OSs, as illustrated in Fig. 1c-d.

The overlapping situations of two OSs are distinguished by whether they have core overlaps (Fig. 1c) or not (Fig. 1d). A sufficient degree of overlap is discriminated by more than 50% for DORs, which is consistent with that in Williams and Houze

(1987). If their cores have overlaps, with the DOR between either cores or OSs greater than 50%, the major parts of those pairs of OSs in situations (i), (ii) and (iv) in Fig. 1b all sufficiently overlap and thus are considered the evolution of the same OS at different times. The situation (iii) in Fig. 1c with DORs of both cores and OSs less than 50% indicates that these two OSs only overlap in margins, with none of associations in time. In Fig. 1d, when the cores of two OSs do not overlap, the determinant of the OS association relies on the DOR between OSs and the DOR of OSs to cores. In those cases, only in situation (ii) in Fig.

1d, with large overlaps of their major parts and those two DOR indices both larger than 50%, the pair of OSs are associated in time. Those pairs of OSs in the other situations in Fig. 1d are obviously not associated. Overall, if the DORs of two OSs satisfy the overlapping conditions of (i), (ii) and (iv) in Fig. 1c and (ii) in Fig. 1d, they are associated in time and regarded as the same OS evolving with time.

In Fig. 1c-d, the occurrence frequency of those overlapping conditions is listed at the top of each subpanel (the blue

numbers), in which the red numbers in the parentheses refer to the frequency for conventional stationary overlaps without consideration of the OS movement. Here, every pair of OSs with overlaps is counted as one sample. For instance, if one OS has overlaps with five OSs of the next moment, there are five pairs of overlapping OSs, and the sample number is five. The frequency is defined as the occurrence of each overlapping condition in Fig. 1c-d divided by the total sample number. It is not surprising that the condition of (iii) in Fig. 1d accounts for the largest portion of samples, since the OSs in CCOs are close to

each other and their margins easily overlap. Compared with the stationary overlaps, the dynamic overlaps increase the frequency of the overlapping condition of (ii) of Fig. 1c twofold and decrease the frequency of all other conditions. Overall, the dynamic overlaps increase the frequency of associations (the sum of frequencies of (i), (ii) and (iv) in Fig. 1c and (ii) in Fig. 1d) by 2.5% from 21.2% (the frequency of associations for stationary overlaps) to 23.7%.

For two temporally associated OSs, the development and decay can be inferred from the variation in the cold-core $BT_{11}$

and area, as shown in Fig. 1e. The variation in the cold-core $BT_{11}$ is considered first before the variation in the cold-core areas for discriminating between development and decay. If the cold-core $BT_{11}$ is colder with time, or if the cold-core $BT_{11}$ is the same but the cold-core area is larger with time, the OS is developing; otherwise, it decays. Notably, OSs are not necessarily associated with only one OS. As shown in Fig. 1e, mergers and splits are allowed in dynamic overlaps and are identified as the many-to-one and one-to-many OS associations, respectively. The merged cold-core $BT_{11}$ and the splitting cold-core $BT_{11}$ are

documented as the $BT_{11}$ of mergers and splits, respectively.

   In comparison, conventional fixed-threshold tracking infers development and decay from the variation in the area, in which the tracking target might be very complicated and involve multiple convective activities at different life stages. However, the variable-$BT_{11}$ tracking infers the development and decay of each partitioned OS from the variation in both the cold-core $BT_{11}$ and the area. Additionally, mergers and splits in fixed-threshold tracking are dependent on the selection of the $BT_{11}$

threshold. Owing to the selection of the fixed $BT_{11}$ threshold, convective systems are usually connected under a warmer threshold but are disconnected under a colder threshold. As illustrated in Fig. 1f, if under the fixed threshold of 260 K, no mergers or splits occur. If under the fixed threshold of 220 K, the cutoff of the CCO by 220 K is the connected complex of multiple cores or two disconnected parts at different times. This change in the connecting conditions over time under the selected fixed threshold results in mergers and splits in fixed-threshold tracking. If under the fixed threshold of 200 K, the

mergers and splits of cold cores are captured. It manifests that mergers and splits in fixed-threshold tracking can be attributed to many reasons: threshold selection, changes in the connecting conditions, and variations in cold cores over time. In contrast, in variable-$BT_{11}$ tracking, mergers and splits are not influenced by changes in the connecting conditions over time but are related only to the variation in cold cores, as illustrated in Fig. 1e-f.

### 3.4  Quality control and validation of variable-$BT_{11}$ segment tracking

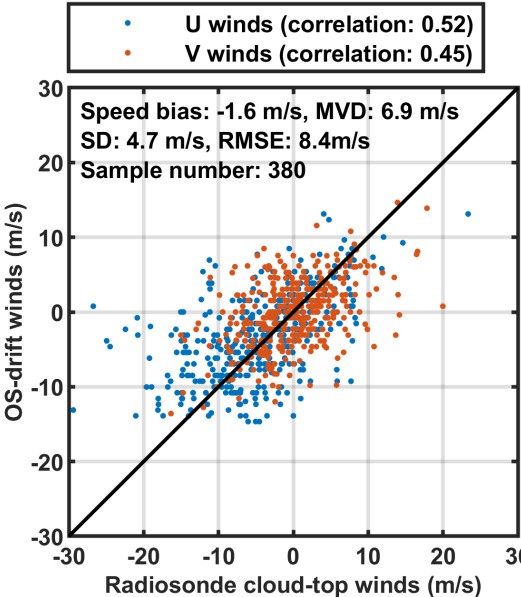


**Figure 3.** Comparisons of the U and V wind speeds between the tracked motions and radiosonde cloud-top winds within 150 km at Darwin, Manus and Nauru in 2006 with a total sample number of 380.

   Quality control for missing images and the OSs touching the image edges is conducted. If the missing time gap between

two satellite images exceeds 2 hours, the OSs in these two images and all life cycles of these OSs are excluded from the analyses. Additionally, the OSs touching the image edges and all life cycles of the OSs touching the edges are excluded.

   There are no direct observations to validate whether OSs are correctly associated. However, some of the tracked behaviors (e.g., the tracked motions of OSs) can be examined against the measurements of other sensors. Only if the tracking is correct would the derived OS-drift winds perform well. Thus, the OS-drift winds are compared against the radiosonde cloud-top winds

at three ARM tropical sites in Darwin, Manus and Nauru. Here, the cloud-top winds are derived by combining the radar and

radiosonde observations at those sites (see more details in Sect. 2.3) as the observational reference to examine the tracked OS motions from the hourly satellite images in 2006. To collocate the observations from the ground-based sites and satellites, the tracked OS-drift winds from the GEO observations that are closest to the time of the cloud-top wind observations and nearest to the site locations are used to compare with the cloud-top winds at those ground-based sites. The observational time difference is no more than one hour and the tracked OS core centroid is within 150 km of those ARM site locations. These settings are consistent with those of previous studies in which the performance of cloud-drift winds was examined (Nieman et al., 1997; Santek et al., 2019; Daniels et al., 2020).

In Fig. 3, the OS-drift winds are significantly correlated with the radiosonde cloud-top winds. The correlations are 0.52 and 0.45 at the 99% significance level for the U and V wind components, respectively. On average, the OS-drift winds are slower than the radiosonde-observed winds are, with a mean bias of -1.6 m/s. A slow speed bias of 1-2 m/s is common for cloud-drift winds (Santek et al., 2019). Owing to the limitations of the spatial and temporal resolutions (5 km and 1 hour, respectively), the least identifiable speed variation is approximately 5 km/hour (1.4 m/s), which is a possible reason for the slow speed bias. The bias in the mean angle is very small (0.5 degrees). The MVD, SD and RMSE are 6.9, 4.7 and 8.3 m/s, respectively. These biases are not surprising since real-world clouds do not strictly flow with ambient winds. In addition, some bias might be attributed to the uncertainty in the cloud-top heights. For its detection, the MMCR might underestimate the cloud top height since its signal would attenuate quickly for deep convective clouds (Hollars et al., 2004). In convective systems, the motion of air is highly organized (Houze, 2004); thus, system movement might be inconsistent with the observed winds at the cloud-top height. Typically, the RMSE of the vector between the cloud-drift winds and the reference cloud-top winds is approximately 6-13 m/s, according to previous studies (Santek et al., 2019; Bresky et al., 2012). This finding indicates that the tracked motions of OSs are reasonable, and thereby variable-$BT_{11}$ segment tracking is appropriate.

### 3.5 Comparison with conventional fixed-threshold tracking

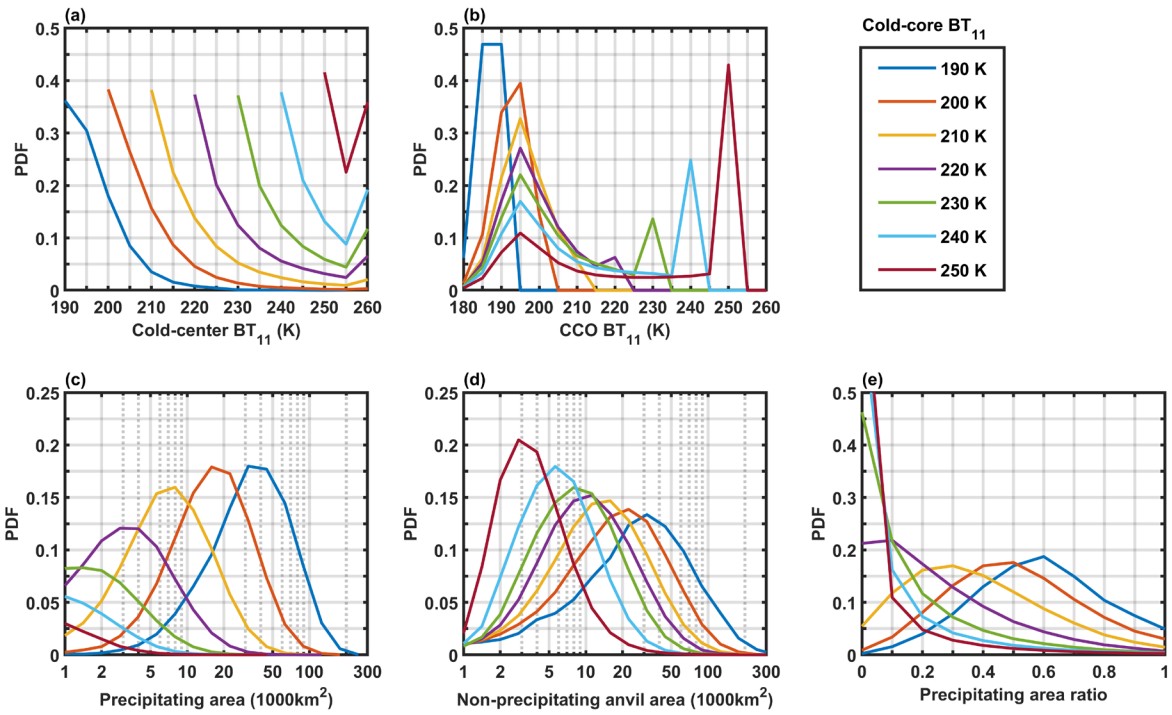

**Figure 4.** The OS structural characteristics of CCOs. PDFs of cold-center $BT_{11}$ (a), CCO $BT_{11}$ (b), the precipitating area (c), the non-precipitating area (d), and the ratio of the precipitating area (e) for the OSs of the cold-core $BT_{11}$ from 190-250 K.

The fundamental difference between fixed-threshold and variable-$BT_{11}$ tracking is target selection. With the fixed threshold of $BT_{11}$, the connected convections of multiple cold cores are identified as tracking targets, and only the area information is accessible for their life cycles. With the OS as tracking targets, variable-$BT_{11}$ tracking is capable of documenting the detailed evolution of each OS within CCOs, such as the developing depth, connecting conditions and

contributions to precipitation and anvil clouds.

The complexity of convective organizations determines the use of fixed-threshold or variable-$BT_{11}$ tracking. Specifically, if the structure of convection is simple with only one cold core, it can be simply tracked via fixed-threshold tracking. Otherwise, for the complex organization of connected convective activities with multiple cold cores, variable-$BT_{11}$ tracking is suitable since it is capable of segmenting CCOs into OSs for tracking.

In Fig. 4, the OS structural characteristics (i.e., the connecting conditions with other surrounding OSs in CCOs and their contributions to precipitation and anvil cloud areas) of different development depths with the cold-core $BT_{11}$ from 190-250 K are investigated. In Fig. 4a, for the OSs of the cold core from 190-250 K, the probability distribution functions (PDFs) of the cold-center $BT_{11}$ are shown. The cold-core and cold-center $BT_{11}$ are both identified by 5-K-interval adaptive thresholds (see details in Section 3.1). The PDFs in Fig. 4a have a maximum peak of approximately 36-41% when the cold-center $BT_{11}$ is

equal to the cold-core $BT_{11}$. This implies that for most of them only the cold core can be isolated by the fixed threshold. For the deep convection of the cold-core $BT_{11}$ at 190-220 K, the isolated structure with a cold-center $BT_{11}$ of 260 K is rare, but it is relatively more frequent and seems to be another mode for the shallow warm systems of the cold-core $BT_{11}$ at 230-260 K. However, fixed-threshold tracking is not capable of discriminating between isolated and complicated structures.

There is no doubt that the warmer the selected $BT_{11}$ threshold is, the more complex the identified target is in the fixed-

threshold identification. However, can one cold $BT_{11}$ threshold be used to avoid complicated connected convective organizations? If feasible, the fixed-$BT_{11}$ tracking under the cold threshold performs well. For instance, Feng et al. (2018) tried to use two thresholds to identify convective systems with a cold threshold of 225 K to capture the cold core and a warm threshold of 241 K to find the cloud pixels associated with the cold cores. In this case, is the 225-K cutoff a simple or complicated structure? If under the fixed threshold of 225 K, Fig. 4a shows that:

(1)   For the OSs of the cold-core $BT_{11}$ from 230-260 K, they would be ignored since these OSs develop warmer than 225 K;
(2)   For the OSs of the cold-core $BT_{11}$ from 190-220 K and the cold-center $BT_{11}$ from 190-220 K, they would be in complicated convective organizations, and cannot be simply identified by the fixed threshold of 225 K;
(3)   For the OSs of the cold-core $BT_{11}$ from 190-220 K and the cold-center $BT_{11}$ from 225-260 K, they can be directly isolated by the fixed threshold of 225 K, but it accounts for only a small portion of the OSs of the cold-core $BT_{11}$ from 190-220 K.

This implies that even under the cold $BT_{11}$ threshold, most of the identified targets still have complex organizations.

Fig. 4b shows the PDFs of CCO $BT_{11}$, which refers to the coldest cold-core $BT_{11}$ in the CCO. The CCO $BT_{11}$ can help to further distinguish the connecting conditions of the OS at different depths of development. If the CCO $BT_{11}$ is colder than the OS cold-core $BT_{11}$, the OS is in a deeper CCO and connected with a colder OS. Otherwise, if the CCO $BT_{11}$ is equal to the OS cold-core $BT_{11}$, the OS is the isolated structure or connected with a warmer OS. For the OSs of the cold-core $BT_{11}$ from 200-

260 K, the PDFs of the CCO $BT_{11}$ all peak at 195 K. It implies that they are the most frequently clustered in the 195-K CCO. For the OSs of cold-core $BT_{11}$ from 230-260 K, another peak of the PDFs of their CCO $BT_{11}$ is at their cold-core $BT_{11}$. Fig. 4a also shows that these OSs of the cold-core $BT_{11}$ from 230-260 K are more likely to have the cold-center $BT_{11}$ occurring at 260 K. This implies that warm-core structures are more likely to be isolated than cold-core structures. As a result, deep convective activities are mostly accompanied by the clustered complex organization, and variable-$BT_{11}$ segment tracking is more suitable

than fixed-threshold tracking for documenting their behaviors.

In variable-$BT_{11}$ tracking, precipitation and anvil clouds can be explicitly attributed to unique cold cores. In Figs. 4c-d, the distributions of the precipitation and anvil areas are lognormal and closely related to those of cold-core $BT_{11}$. The colder the cold-core $BT_{11}$ is, the greater the precipitation and anvil areas to which the OS contributes. In Fig. 4e, the ratio of the OS precipitation area to the whole OS area is inversely proportional to the cold-core $BT_{11}$. The OSs of colder cores are dominated

by more precipitation, and these OSs still contribute to more anvil clouds than the OSs of warmer cores. Similar to Lindzen et al. (2001), the ratio of the OS precipitation area to the whole OS area can also be understood as a diagnostic of the precipitation efficiency or detrainment effect. In Figs. 4c-e, the observed relationship between the $BT_{11}$ structure and precipitation efficiency

might be expected. Storms with higher precipitation efficiency generally have less dry air entrainment, which may allow updrafts to reach higher altitudes and lower $BT_{11}$.

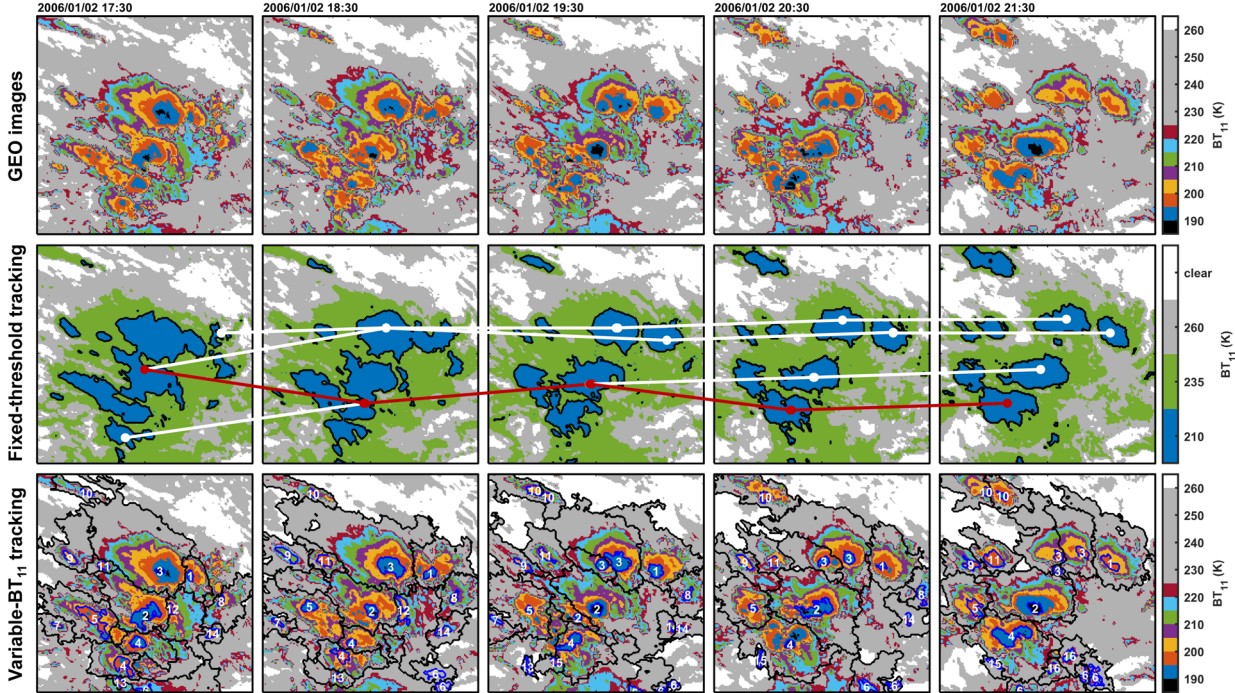

**Figure 5.** Examples illustrating the difference between conventional fixed-threshold tracking and novel variable-$BT_{11}$ segment tracking. Uppermost panel: GEO $BT_{11}$ images taken between 5°S-15°S and 120°E-130°E from January 2, 17:30 to 21:30 UTC in 2006. Middle panel: the tracked life cycles based on the fixed thresholds of 210, 235 and 260 K. The white lines represent the tree of the tracked life cycle under the fixed threshold of 210 K and the red lines represent the major branch obtained by selecting the largest area. Bottom panels: variable-$BT_{11}$ identification and segment tracking. In the bottom panels, the blue contours indicate the cold cores, the black contours are the OSs in the CCOs, and the number at the core centroids indicates the identification of life cycles.

Examples of the conventional fixed-threshold and novel variable-$BT_{11}$ tracking algorithms are shown in Fig. 5 to illustrate their differences. From the GEO images in the uppermost panel of Fig. 5, visually, those convective activities are connected in CCOs but have distinct behaviors to decay, split, develop and merge over time. In the middle panel of Fig. 5, these behaviors of connected convections are barely distinguished by fixed-threshold tracking. With thresholds of 235 K and 260 K, the whole complex organization of connected convections is identified as the tracking target. By the cold threshold of 210 K, only a small part of the CCO is identified; nevertheless, those connected convections of distinct behaviors are still poorly distinguished. Additionally, in fixed-threshold tracking at 210 K, mergers and splits are caused by variations in whether convections are connected under the 210-K threshold. In this case, the tree of the tracked life cycle is too complicated to analyze and is usually simplified by focusing only on the largest area at different times as the major branch of the life cycle. The major branch (the red line in the middle panel of Fig. 5) begins with the large complex organization of connected convections but ends with only one of disconnected parts. Only the area information is available to describe the life cycle in fixed-threshold tracking. In comparison, in the bottom panel of Fig. 5, on the basis of adaptive variable-$BT_{11}$ identification and segment tracking, these connected convective activities are separately tracked as the decaying and splitting No. 3 OS, developing No. 2 OS, merging and developing No. 4 OS, etc. The mergers and splits of OSs are well tracked and not influenced by the variations in the connecting conditions over time. The area and cold-core $BT_{11}$ information are both available to describe the tracked life cycles.

## 4. Relationships of precipitation and anvil production with the structural evolution of $BT_{11}$ for oceanic convection over the tropical western Pacific Ocean

The warm pool of the tropical western Pacific Ocean (130°W-170°E, 20°S-20°N) is a typical region of oceanic convection that precipitates and produces anvil clouds (Wall et al., 2018). In this section, only the CCOs and OSs over the oceans in this

region are considered for investigating the behaviors of the oceanic convection precipitating and producing anvil clouds. Notably, the anvil identification requires that the $BT_{11}$ is colder than 260 K and the precipitation is less than 1 mm/hour. It can be used to reflect the anvil productivity in the convective systems (Yuan and Houze, 2010; Yuan et al., 2011; Yuan and Houze, 2013), but much the area of detained cirrus has the $BT_{11}$ warmer than 260 K in reality (Gasparini et al., 2022; Sokol and Hartmann, 2020; Berry and Mace, 2014). Normally, those thin cirrus clouds are not well identified by GEO radiometers and thus in this work, the anvil just refers to the thick anvil portion identified by the 260-K $BT_{11}$ threshold but not all detrained anvil cirrus clouds.

The total precipitation and anvil cloud amounts of convection are important for tropical water and radiative budgets. They can be attributed to two factors: (1) the occurrence frequency of convection and (2) the precipitation and anvil production for the duration of convection. However, over the warm pool of tropical oceans, convective activities are clustered in CCOs and their precipitation and produced anvil clouds are merged (as discussed in Sect. 3). As a result, identifying their contributions to precipitation and anvil clouds is difficult. On the other hand, the CCO is a large cluster for a series of alternating successive convective activities, which are initiated at different times and evolve in different ways. Thus, there is a dilemma in tracking convection: convection is not isolated naturally for tracking, whereas the CCO is the envelope of many convections whose precipitating and anvil areas are mixed, and it is difficult to identify single convective processes from the CCO life cycle.

It has long been well observed by various active and passive sensors that tropical convections have core structures, e.g., convective pillars observed by active sensors (Igel et al., 2014; Takahashi and Luo, 2012; Deng et al., 2016), heavy raining cores observed by radar or passive microwave radiometer (Yuan and Houze, 2010; Feng et al., 2011), and cold cores of $BT_{11}$ observed by GEO or MODIS radiometers (Yuan and Houze, 2010; Yang et al., 2023; La and Messager, 2021). Although convective structures can be better identified by active sensors than by passive sensors, active sensors are only available at a limited number of ground-based sites or on polar-orbit satellites, and their samplings are too sparse for tracking. Yuan and Houze (2010) and Yang et al. (2023) both used active and passive sensors in combination and demonstrated that the $BT_{11}$ structures are strongly associated with the convective structures. Yuan and Houze (2010) reported that cold and warm $BT_{11}$ correspond to two distinct types of clouds detected by active sensors: very deep convective clouds and elevated anvil clouds. They partitioned the CCO into single-core high cloud systems (i.e., the OS defined in this work) and identified those OSs with heavy precipitation and the cold-core $BT_{11}$ colder than 220 K as mesoscale convective systems (MCSs). For these MCSs, Yuan et al. (2011) observed that the cloud vertical structures are well organized, in which high-topped clouds extend outward from raining cores and the thickness of the anvil and the sizes of ice particles are closely related to the distance to the raining cores. Similarly, Yang et al. (2023) identified the cold cores of $BT_{11}$ as convective centers and also found that the cold-core structures of $BT_{11}$ are highly consistent with the convective structures detected by active sensors. These findings suggest that cold cores of $BT_{11}$ can be used to identify the most convective-developing centers and distinguish convective activities in CCOs. In Sect. 3, a novel algorithm was developed to accomplish tracking for convective cold-core structures on the basis of previous studies (Yuan and Houze, 2010; Yuan et al., 2011) and GEO observations.

These OSs can be used to infer different convective activities clustered in CCOs. They are organized differently with various depths of development, precipitation and anvil production and have distinct evolution processes. In this section, on the basis of variable-$BT_{11}$ tracking, the relationships of convective contributions to precipitation and anvil clouds with their $BT_{11}$ structural evolution are explored. This would provide an opportunity to compare convections of different development strengths and evolutions for their contributions to precipitation and anvil clouds.

**4.1 Relationships of lifetime-accumulated precipitation and anvil clouds with cold-core-peak BT$_{11}$**

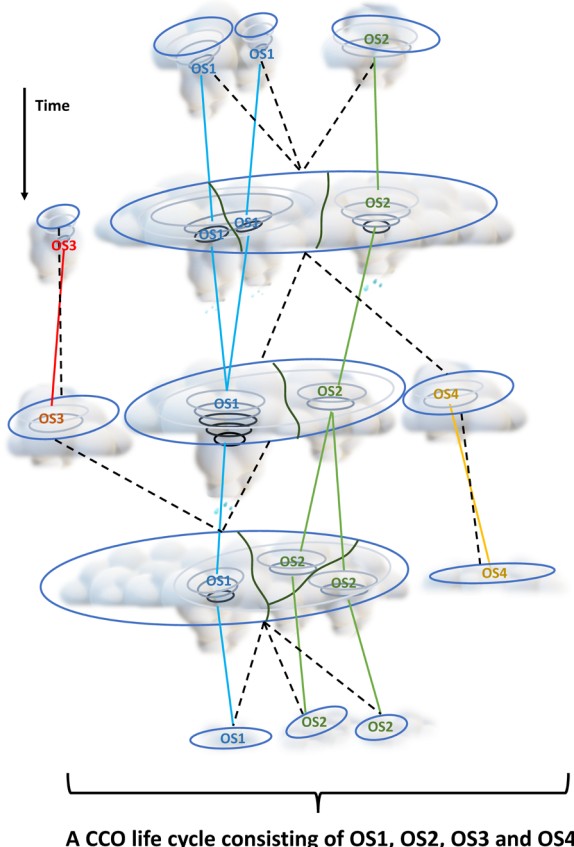

**A CCO life cycle consisting of OS1, OS2, OS3 and OS4**

Figure 6. Illustrations of the difference between tracking a CCO and tracking OSs. The CCO life cycle include four OS life cycles. The dash black line indicates the tree of the CCO life cycle. The blue, green, red and yellow lines indicate the tree of OS1, OS2, OS3 and OS4 life cycles, respectively.

Fig. 6 illustrates an idealized tracking for a CCO and its OSs. The real-world CCO tracking can be much longer and more complicated than that in Fig. 6, and here, it is just used to illustrate how to understand the CCO and OS tracking. As illustrated in Fig. 6, the CCO tracking (dashed black line) can capture the variation in precipitation and anvil areas contributed by multiple convections, but it does not link these variations to specific convections. Mergers and splits in the CCO life cycle reflect the connections and disconnections between different convections. With the OS tracking, the CCO life cycle can be decomposed into the life cycles of its structural components (the colored lines in Fig. 6). It can be recognized that the life cycle of the CCO starts with three convective activities, and with time two of them are merged into the OS1 life cycle and the left one splits into two as the OS2 life cycle. In this way, precipitation and anvil clouds are associated with convective activities in CCOs. On the other hand, the CCO is a large envelope of many convective activities and it is not expected that they all have simple perfect life cycles from convective initiation to anvil dissipation. The OS might just be born from the split of the anvil or the secondary convective activity in its parent stronger convective body (e.g., the OS4 life cycle in Fig. 6) and ends by merging into the anvil in the CCO (e.g., the OS3 life cycle in Fig. 6). The OS tracking documents the life cycle of the core structure from initiation to dissipation. It can be expected that the active convective activities have robust and durable core structures in CCOs, while weak secondary convective activities are fragile and short-lived. In Fig. 7 and Fig. 8, the basic features of OS life cycles of different peaking strengths are investigated for their occurrence, duration and contributions to precipitation and anvil clouds.

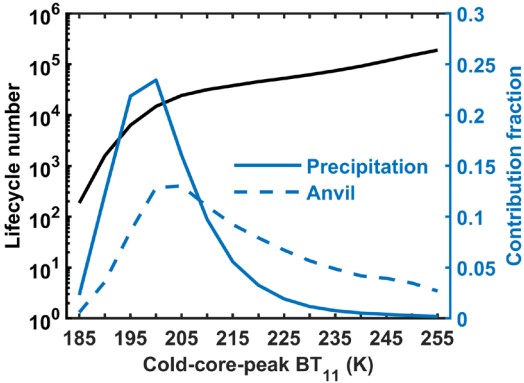

**Figure 7.** Sample numbers of tracked OS life cycles with cold-core-peak $BT_{11}$ values from 185-255 K in the tropical western Pacific (130°W-170°E, 20°S-20°N) in 2006. The contribution fractions of the OS life cycles to the precipitation and anvil cloud amounts are shown on the right Y-axis.

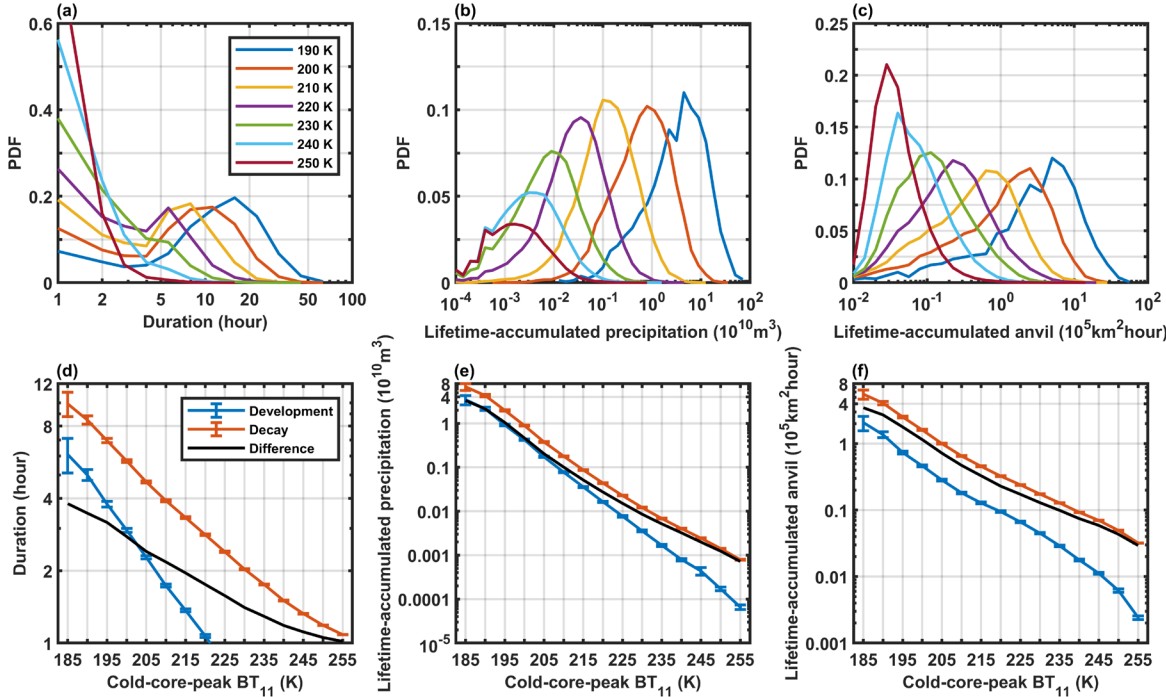

**Figure 8.** PDFs of the duration (a), precipitation (b) and non-precipitating anvil amount (c) of lifetime accumulation, for the OS life cycles of different cold-core-peak $BT_{11}$ values from 190-250 K. The composites of the duration (d) and the lifetime-accumulated precipitation (e) and anvil amount (f) contributed by the development (blue lines) and decay stages (red lines) as a function of the cold-core-peak $BT_{11}$ from 185-255 K. The black lines represent the differences in the duration and the lifetime-accumulated precipitation and anvil between the development and decay stages in (d-f), respectively. The error bars indicate the 95% confidence intervals of the means based on the t test.

For OS life cycles, the cold-core-peak $BT_{11}$ is identified as the coldest cold-core $BT_{11}$ in life cycles and is used to represent the convective peaking strength. The OS life cycles are classified by the cold-core-peak $BT_{11}$. In Fig. 7, the sample numbers of life cycles of different cold-core-peak $BT_{11}$ values over the tropical western Pacific Ocean in 2006 are shown. The warmer the cold-core-peak $BT_{11}$ is, the greater the number of life cycles is. The life cycle of the cold-core-peak $BT_{11}$ at 185 K has only over one hundred samples, and the life cycle of the cold-core-peak $BT_{11}$ at 255 K has hundreds of thousands of samples in one year. In addition, Fig. 7 also shows the fractions of the contributions of OS life cycles of different cold-core-peak $BT_{11}$ values to the total precipitation and anvil cloud amounts. Here, the fraction of contribution refers to the sum of the precipitation (anvil) produced by all OS life cycles in each bin of the cold-core-peak $BT_{11}$ divided by the total precipitation (anvil). The OS life cycles of the cold-core-peak $BT_{11}$ at 200 K have the largest contribution to both the precipitation and anvil cloud amounts. Although the life cycles of the cold-core-peak $BT_{11}$ values warmer than 235 K have a great number of samples, those life

cycles contribute to only no more than 5% of the total precipitation and anvil cloud amounts. Nevertheless, those warm structures seem to be more important for anvil clouds than for precipitation.

In Figs. 8a-c, for different cold-core-peak $BT_{11}$ values, the PDFs of the OS duration and the OS lifetime-accumulated precipitation and anvil amount are shown. Here, the OS duration refers to the time of the tracked OS life cycle. The OS lifetime-accumulated precipitation and anvil amount are the sum of the observed OS precipitation and anvil in hourly satellite images during its lifetime. For example, for the OS1 life cycle in Fig. 6, the lifetime-accumulated precipitation and anvil amount are the sum of the hourly precipitation and anvil in the OS1 life tree.

In Fig. 8a, for the duration of the OS with the cold-core-peak $BT_{11}$ that is warmer than 220 K, the PDFs peak at 1 hour and most of the durations are less than 5 hours. For the life cycle of the peak $BT_{11}$ colder than 220 K, the OS duration has two modes: short-lived (1-4 hours) and long-lived (at least 5 hours). As discussed previously, it is not expected that convective activities in CCOs all have simple perfect life cycles from convective initiation to anvil dissipation. Some OSs, even for those very cold structures, are just very short-lived overshooting with only a 1-hour duration for the secondary convective activity in its parent stronger convective body and then disappear or may be annexed by its surrounding stronger vertical-developing convection in CCOs.

In Figs. 8b-c, the PDFs of the accumulated precipitation and anvil amounts contributed by the OS life cycles basically conform to the lognormal distribution. Overall, the duration, precipitation and anvil amount of lifetime accumulation are inversely proportional to the cold-core-peak $BT_{11}$.

In Figs. 8d-f, the relationships of the OS duration and the OS lifetime-accumulated precipitation and anvil amount with the cold-core-peak $BT_{11}$ are further investigated. The OS life cycle is separated into development and decay stages. The development (decay) stage is defined as the stage before (after) the cold core peaks at the coldest $BT_{11}$ with the largest core area. The peak is counted as the development stage. Thus, for short-lived OSs with a duration of only 1 hour, their development time is just 1 hour and the decay time is zero. In Fig. 8d, only the OS life cycles of the cold-core-peak $BT_{11}$ colder than 220 K have a development of more than 1 hour, whereas the OS life cycles of the peak $BT_{11}$ that is warmer than 220 K directly decay and disappear rapidly within a few hours. This implies that most of these warm structures are just the anvil split with weak secondary convection from its main convective body (such as the OS4 in Fig. 6). On the basis of the PDFs of the OS duration (Fig. 8a) and the duration of development and decay (Fig. 8d), those OSs with the cold-core-peak $BT_{11}$ colder than 220 K and the durations greater than 5 hours are identified as active convective activities; otherwise, the OSs with warmer or short-lived core structures are weak secondary convective activities.

Figs. 8d-f show that the process of decay is longer than the process of development and that more precipitation and anvil clouds are contributed by the process of decay than by the process of development. Overall, the OS duration and the lifetime-accumulated precipitation and anvil amount have simple loglinear relationships with the peak $BT_{11}$ in both the development and decay stages. The difference in the duration and the accumulated precipitation and anvil cloud amounts between the two stages (the black line in Figs. 8d-f) also exponentially increases, with the core peaking at colder $BT_{11}$ values. The duration of the development and decay processes and the two key components of the convective cloud water budget, i.e., the lifetime-accumulated precipitation and anvil cloud amounts, are closely related to the peaking cold-core structures.

## 4.2 Influence of mergers and splits on precipitation and anvil production

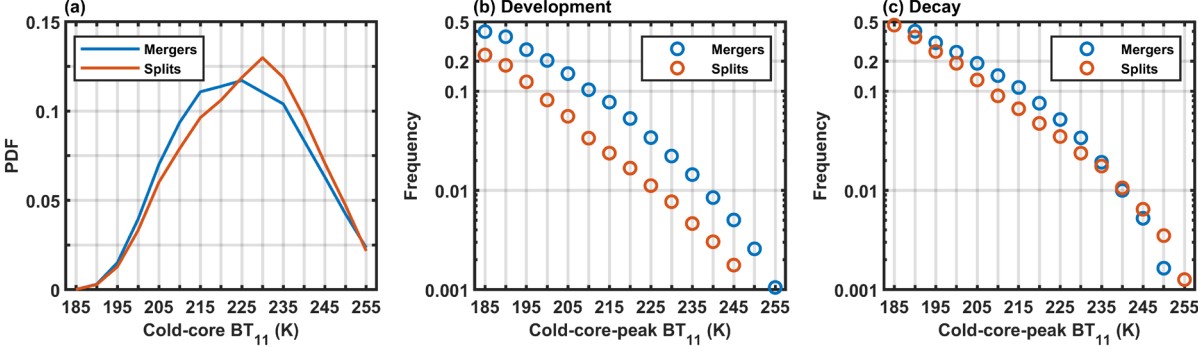

**Figure 9.** (a) PDFs of $BT_{11}$ values for mergers (blue lines) and splits (red lines). The occurrence frequency of mergers and splits in the life cycles of the cold-core-peak $BT_{11}$ from 185-255 K in the development (b) and decay (c) stages.

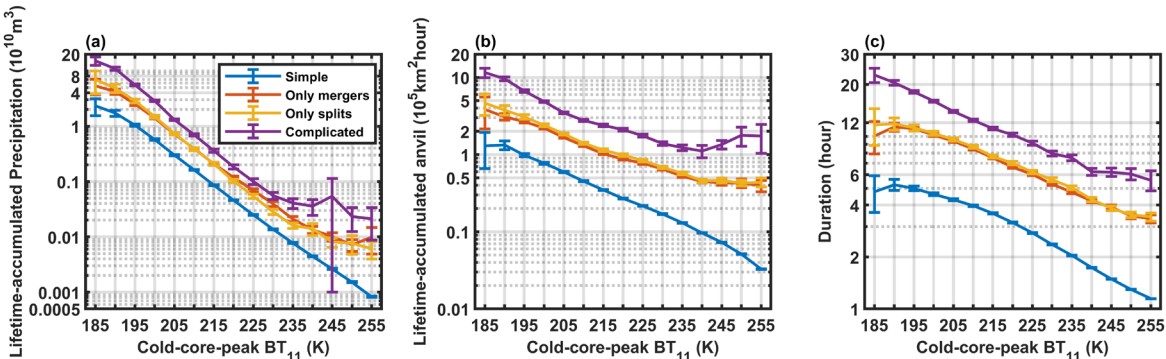

**Figure 10.** Composites of the lifetime-accumulated precipitation (a), anvil cloud amounts (b), and durations (c) of different lifecycle types in each bin of the cold-core-peak $BT_{11}$ from 185-255 K. The blue, red, yellow and purple lines indicate the simple, only-merger, only-split and complicated life cycles, respectively. The error bars indicate the 95% confidence intervals of the means based on the t test.

Mergers and splits represent activities of cold cores in the OS tracking. In Fig. 9a, according to the PDFs of the cold-core $BT_{11}$ of mergers and splits, the cold-core $BT_{11}$ of mergers is distributed at colder $BT_{11}$ values than that of splits. This implies that mergers are more likely to occur for cold structures, whereas splits are more likely to occur for warm structures. Figs. 9b-c show the occurrence frequency of mergers and splits in the OS life cycles of different cold-core-peak $BT_{11}$ values, in the development and decay stages, respectively. It is somewhat surprising that the frequency of mergers and splits still has a loglinear relationship with the cold-core-peak $BT_{11}$. In the development process, mergers are more likely to occur than splits. In the decay process, mergers and splits have similar occurrence frequencies, but the splits in the decay process are more frequent than those in the development process.

According to the occurrence of mergers and splits, OS life cycles can be further classified into simple (no mergers or splits), only-merger, only-split and complicated (both mergers and splits) types. In Figs. 10a-c, the lifetime-accumulated precipitation, anvil cloud amount and duration are strongly related to the occurrence of mergers and splits. For the same cold-core-peak $BT_{11}$, the complicated life cycles have the largest accumulated precipitation, anvil and duration among all types of life cycles. The only-merger and only-split life cycles have similar accumulated precipitation, anvil and duration values that are greater than those of the simple life cycles. Interestingly, for different types of life cycles, the slopes of the loglinear relationships of the lifetime-accumulated precipitation, anvil and duration with the cold-core-peak $BT_{11}$ are nearly invariant. This implies that mergers and splits do not influence the dependence of the lifetime-accumulated precipitation and anvil cloud amounts on the $BT_{11}$ structures and that the increased precipitation and anvil cloud amounts caused by the mergers and splits also conform to a loglinear relationship with the cold-core-peak $BT_{11}$.

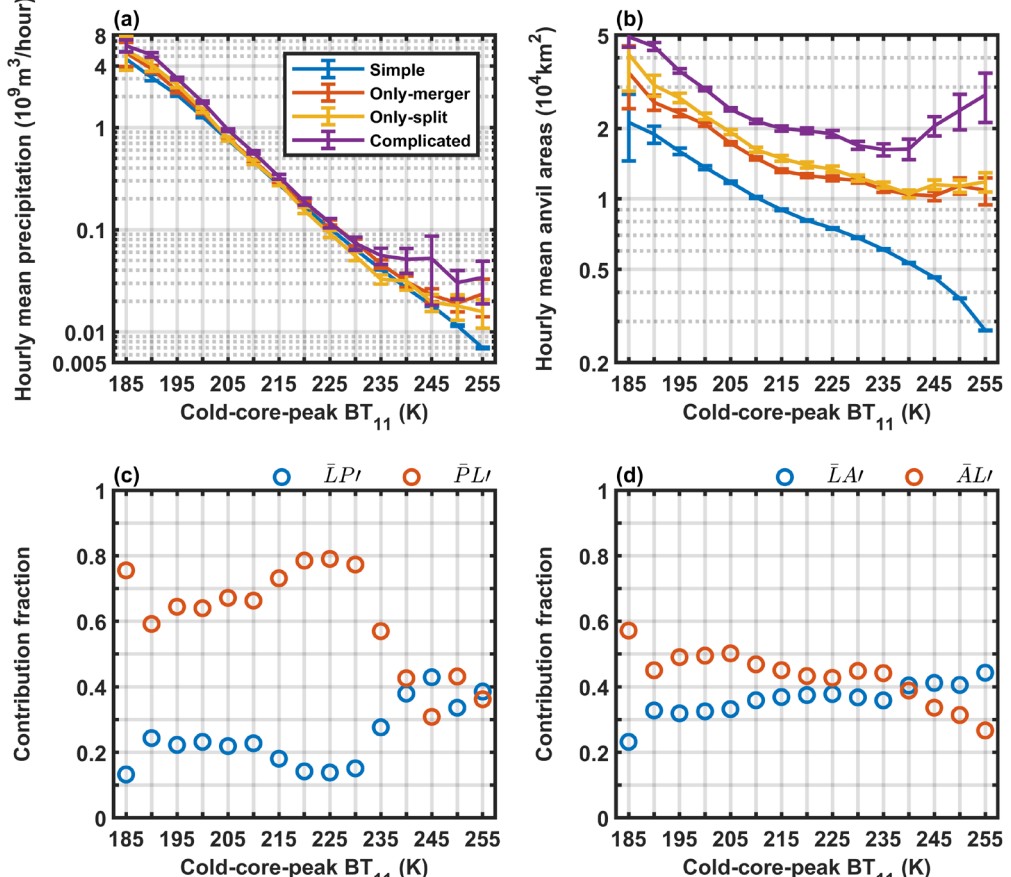

**Figure 11.** Composites of the hourly mean precipitation (a) and anvil cloud amounts (b) of different lifecycle types in each bin of the cold-core-peak $BT_{11}$. The blue, red, yellow and purple lines indicate the simple, only-merger, only-split and complicated life cycles, respectively. (c) The fractions of contributions of the hourly precipitation anomalies ($\bar{L}P'$) and lifetime anomalies ($\bar{P}L'$) to the variation in lifetime-accumulated precipitation. (d) The fractions of contributions of the hourly anvil production anomalies ($\bar{L}A'$) and lifetime anomalies ($\bar{A}L'$) to the variation in the lifetime-accumulated anvil amount. The error bars indicate the 95% confidence intervals of the means based on the t test.

How do mergers and splits influence the lifetime-accumulated precipitation and anvil cloud amounts? This question is simply explored from the OS tracking. In the OS life cycle, the variation in the accumulated precipitation and anvil cloud amounts can be attributed to two possible factors: (1) the hourly precipitation and anvil production in the life cycle are enhanced by mergers and splits, and (2) the lifetime is prolonged by mergers and splits.

In Figs. 11a-b, the hourly mean precipitation and anvil amount in the OS life cycles are shown for different types of life cycles. For the same cold-core-peak $BT_{11}$, the hourly mean precipitation of different lifecycle types is nearly invariant (Fig. 11a). However, in the life cycles with the occurrence of mergers and splits, the hourly mean anvil production is enhanced (Fig. 11b), and the lifetime (L) is significantly prolonged (Fig. 10c). To quantify their impacts, in Figs. 11c-d, the anomalies of the lifetime-accumulated precipitation and anvil cloud amounts can be decomposed as follows:

$$PL - \bar{P}\bar{L} = \bar{L}P' + \bar{P}L' + P'L', \qquad (7)$$

$$AL - \bar{A}\bar{L} = \bar{L}A' + \bar{A}L' + A'L'. \qquad (8)$$

P and A are the hourly precipitation and anvil cloud amount, respectively. L is the lifetime. Thus, PL and AL represent the lifetime-accumulated precipitation and anvil cloud amount, respectively. The bar over the letter represents the mean of different lifecycle types. The prime over the letter represents the anomaly of different lifecycle types relative to their mean value. In this way, $\bar{L}P'$ and $\bar{P}L'$ indicate the contributions of the hourly precipitation anomaly and the lifetime anomaly, respectively, to the variation in lifetime-accumulated precipitation. Similarly, $\bar{L}A'$ and $\bar{A}L'$ indicate the contributions of the hourly anvil production and lifetime anomalies, respectively, to the variation in the lifetime-accumulated anvil amount. $P'L'$ and $A'L'$ are high-order small quantities and are neglected. The fraction of the contribution can be computed by dividing the left-hand-side quantities

of Eq. 7 and Eq. 8. Fig. 11c (Fig. 11d) shows the fractions of the contributions of $\bar{L}P'$ and $\bar{P}L'$ ($\bar{L}A'$ and $\bar{A}L'$) to the increase in lifetime-accumulated precipitation (anvil) from simple to complicated life cycles. For the life cycles of the cold-core-peak $BT_{11}$ colder than 220 K, $\bar{L}P'$ has a relatively small contribution of approximately 10-25%, whereas $\bar{P}L'$ has a large contribution of approximately 60-80%. In addition, $\bar{L}A'$ and $\bar{A}L'$ both have positive comparable contribution fractions, approximately 20-40% and 40-60%, respectively. For the warmer life cycles, the contributions from $\bar{L}P'$ and $\bar{L}A'$ increase and are more important than the lifetime anomaly for the variation in the lifetime-accumulated precipitation and anvil cloud amounts.

On average, in comparison with simple life cycles, mergers and splits can significantly prolong the duration of OSs while enhancing the hourly precipitation slightly and increasing the hourly anvil production strongly. From simple to complicated life cycles, a prolonged lifetime accounts for the largest contribution to the increase in accumulated precipitation and anvil clouds for cold structures.

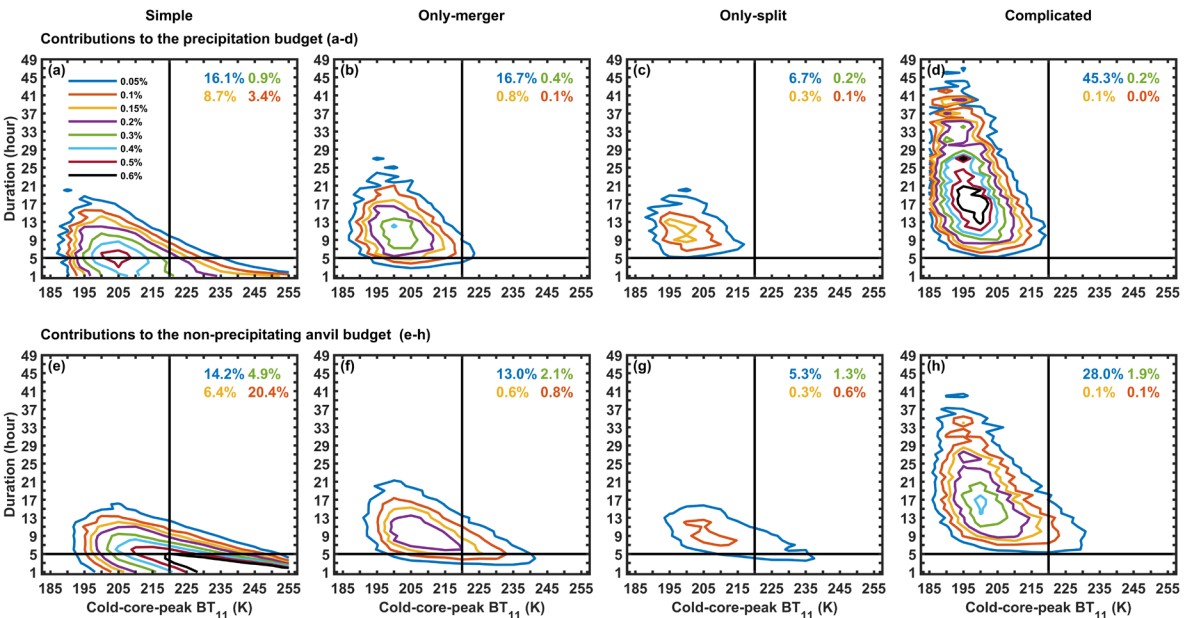

**Figure 12.** Fractions of the contributions of the OS life cycles of different cold-core-peak $BT_{11}$ values and durations to the total precipitation (a-d) and anvil cloud amount (e-h) over the tropical western Pacific Ocean, for simple, only-merger, only-split and complicated life cycles, respectively. Two black straight lines for the peak $BT_{11}$ of 220 K and the duration of 5 hours are used to discriminate between cold and warm and between long-lived and short-lived life cycles, respectively. In each panel, the fractions of contributions of the cold long-lived (blue), warm long-lived (green), cold short-lived (yellow) and warm short-lived (red) are listed in the top right-hand corner.

### 4.3 Contributions of the OS life cycles to the total precipitation and anvil cloud amount

Figs. 8-11 show that the OS duration and the lifetime-accumulated precipitation and anvil cloud amounts all have loglinear relationships with the cold-core-peak $BT_{11}$ and are positively related to the occurrence of mergers and splits. However, the life cycles of cold structures are less common than the life cycles of warm structures and thus might make only small contributions to the total precipitation and anvil clouds (Fig. 7). In Fig. 12, the contributions of the OS life cycles of different cold-core-peak $BT_{11}$ values and durations to the total precipitation and anvil cloud amounts are shown.

The simple life cycle without mergers and splits is the most frequent and accounts for 93.9% of samples. The only-merger, only-split and complicated life cycle have the frequency of only 3.0%, 1.4% and 1.7%, respectively. However, these life cycles with mergers and splits can contribute to a large portion of total precipitation and anvil clouds. For precipitation, as shown in Figs. 12a-d, the complicated life cycles have the largest contribution of 45.6%, compared with the simple (29.1%), only-merger (18%) and only-split (7.3%) life cycles. The cold long-lived life cycles (active convective activities) with the cold-core-peak $BT_{11}$ colder than 220 K and durations of at least 5 hours contribute to 84.8% of the total precipitation, in which the life cycles with mergers and splits contribute to the largest portion (68.7%). This implies that the cold long-lived life cycles with merger and split activities are the most important for precipitation.

For the non-precipitating anvil clouds in Figs. 12e-h, the cold long-lived life cycles still have the largest contribution fraction (60.5%), in which those life cycles with mergers and splits contribute to 46.3 % of the total anvil areas. It implies that active convective activities and the behaviors of mergers and splits are still the most important for the anvil budget. On the other hand, the warm short-lived (warmer than 220 K and less than 5 hours) simple life cycles still account for a relatively large portion of anvil areas (20.4%). These warm and fragile core structures are not efficient at producing anvil clouds but are very frequent in observations and thus also lead to an important contribution to the anvil budget.

## 5. Conclusion

Tropical convection organizations are normally connected complexes of many convective activities. In this work, a novel variable-$BT_{11}$ segment tracking algorithm is established to segment the CCO into OSs for tracking. The tracked motions of OSs are compared against the observational cloud-top winds to examine the rationality of tracking. Strong correlations between the tracked motions and real winds are found, with small differences in the mean speeds (-1.6 m/s) and angles (0.5°). These results confirm that tracking is appropriate.

Compared with the previous fixed-threshold and variable-$BT_{11}$ tracking algorithms that focus only on the variation in the area, the novel tracking algorithm developed in this work is designed to track the core structure from initiation to dissipation under the background of CCOs and to document the evolution of both the area and $BT_{11}$ structure (i.e., the cold-core and cold-center $BT_{11}$ for indicating the developing and connecting conditions, respectively, and mergers and splits of cold cores). Precipitation and anvil clouds are explicitly associated with unique cold cores, which is helpful for quantifications of precipitation and anvil production from different convective activities. The life cycles are described by the activities of cold cores, e.g., cold-core-peak strength, mergers and splits.

The essential difference between the fixed-threshold and novel variable-$BT_{11}$ tracking algorithms is the selection of tracking targets. The complex organizations of multiple cold cores are very frequent and isolated convective bodies are rare, particularly for the deep convection of the cold-core $BT_{11}$ from 190-220 K. This implies that most of the targets identified by the fixed $BT_{11}$ threshold are complex organizations and their segmentation in the variable-$BT_{11}$ tracking algorithm is necessary. Additionally, in fixed-threshold tracking, mergers and splits are caused by many factors: threshold selection, changes in the connecting conditions, and activities of cold cores. This makes the final life cycle in fixed-threshold tracking very complicated. In contrast, in the novel variable-$BT_{11}$ tracking algorithm, mergers and splits are related only to activities of cold cores.

Based on the novel variable-$BT_{11}$ tracking algorithm, the convective processes of precipitation and the production of anvil clouds over the tropical western Pacific Ocean are investigated. Interestingly, the accumulated duration, precipitation and anvil cloud amounts produced in the OS life cycles have simple loglinear relationships with the $BT_{11}$ of the cold core peak. During the life cycle, decay spends more time than development does. More precipitation and anvil cloud amounts are also contributed by the decay stage than by the development stage. The difference in the accumulated duration, precipitation and anvil production between the two stages increases exponentially with the decreasing cold-core-peak $BT_{11}$.

The CCO is the envelope of many convective activities and it is not expected that all of core structures in CCOs have simple perfect life cycles from convective initiation to anvil dissipation. According to the two modes in the PDFs of durations, those cold structures of the cold-core-peak $BT_{11}$ colder than 220 K and durations greater than 5 hours are identified as active convective activities; otherwise, those relatively warm and short-lived structures are secondary weak convective activities.

The occurrence of mergers and splits also strongly relies on the $BT_{11}$ of the cold core peak, still with a loglinear relationship. Overall, mergers are more frequent than splits in the development stage. In the decay stage, the frequency of mergers and splits shows little difference. But, the frequency of splits in the decay stage is greater than that in the development stage. Lifetime-accumulated precipitation and anvil production are positively related to the occurrence of mergers and splits. For the same cold-core-peak $BT_{11}$, the increase in lifetime-accumulated precipitation between the simple and complicated life cycles is mostly attributed to the prolonged lifetime. The increase in the lifetime-accumulated anvil cloud amount between the

simple and complicated life cycles should be attributed to the increase in both the hourly anvil production and the prolonged lifetime. The slope of the loglinear relationship between the lifetime-accumulated precipitation or anvil and the cold-core-peak $BT_{11}$ is almost invariant for different types of life cycles.

For the total cloud water budget over the tropical western Pacific Ocean, most of the total precipitation is contributed by long-lived life cycles with the occurrence of mergers and splits, whereas short-lived simple life cycles are less important for precipitation. However, for the anvil cloud amount, the long-lived complicated life cycle and the short-lived simple life cycle are both important.

## Acknowledgment

This work was supported by the NSFC-41875004 and the National Key R&D Program of China (2016YFC0202000).

## Author contribution

ZW and JY designed the algorithm. ZW carried out the experiments and prepared the manuscript.

## Data and code availability

All data used in this study are available online. The GEO images (Nasa/Larc/Sd/Asdc, 2017) are obtained from the National Aeronautics and Space Administration (NASA) Langley Research Center Atmospheric Science Data Center (https://search.earthdata.nasa.gov/). GPM (Huffman, 2023) is obtained from the Goddard Earth Sciences Data and Information Services Center (GES DISC). The ground-based cloud (S. Giangrande, 1999) and wind (E. Keeler, 2001) observations at the ground-based sites are obtained from the Atmospheric Radiation Measurement user facility, a U.S. Department Of Energy (DOE) office of science user facility managed by the biological and environmental research program (https://www.arm.gov). The code of the anvil tracking algorithm is available upon request.

## Competing interests

The author declares no conflict of interest.

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
