# Peer review of "Observing convective activities in complex convective organizations and their contributions to precipitation and anvil cloud amounts"

_EGUsphere, 2024_

## Author Comment (AC1)

**Response to Anonymous Referee #1**

*Referee #1: The paper by Zhenquan Wang presents a new tracking algorithm for tropical convective systems and uses the algorithm to answer a few science questions about convective storms. Most of the paper is devoted to the tracking algorithm, in which variable brightness temperature (BT11) thresholds are used to identify cloud systems, segment them into convective cores and anvil clouds, and track the evolution, merging, and splitting of the segments over time. One of the main results is that colder BT11 is associated with a greater frequency of mergers and splits. In the last part of the paper, the algorithm is used to examine cloud lifetime, precipitation, and anvil cloud area. These properties tend to display log-linear relationships when plotted against BT11.*

*This is an interesting study and reflects an impressive amount of work by the author. I have no doubt that the tracking algorithm developed here is well motivated and well executed, and it seems like it could produce an interesting dataset from which many questions about convective cloud systems could be examined.*

*However, there are serious issues regarding the clarity of presentation in this paper. I found much of the writing and descriptions of the methodology to be very unclear, and the terminology used for the tracking algorithm was confusing and difficult to grasp. For these reasons, I do not feel equipped to evaluate the appropriateness of the methodology or to understand what the scientific conclusions really mean. So, please excuse me for being unable to provide much constructive feedback here. I would be happy to do so in the future once the presentation has been clarified. Some general comments are below, followed by line comments.*

**Response:** We thank the anonymous reviewer for his/her efforts of reviewing our manuscript. We are very grateful for his/her valuable comments to help us improve the representation of the results. We have carefully taken these comments into account and accordingly revised and reorganized the manuscript to clarify the writing and descriptions of the methodology.

*1. Unclear terminology. Cold-core, cold-center, segmentations, HCSs, organizations, organization segments, mergers & splits. Some of these terms are more clearly defined than others, but the precise meanings need to be clarified (especially HCS). Fig 1a was helpful for understanding centers vs cores...perhaps a similar schematic would help for the other terms.*

**Response:** The definitions of these terminologies have been specifically clarified in the main text. Figure 1 has been revised by adding more cartoon subfigures to help to understand these terminologies and the steps of establishing the variable-BT11 tracking algorithm (as shown below). Table 1 has been added to summarize these terminology definitions for easily checking (as shown below).

The terminology definitions can be checked as follows for how they have been revised in the manuscript:

(1) Terminology used in the target identification:

Complex convective organizations (CCOs): the contiguous area of the BT11 colder than 260 K.

Organization segments (OSs): the OS and the high cloud system (HCS) have exactly the same meaning and both represent the CCO segment of a single cold core, which is the target of the variable-BT11 tracking. It represents the structural component of CCOs. To avoid misunderstanding, in the revised manuscript, only the terminology of "OS" is used.

Cold cores: the local coldest BT11 contour within the OS. The cold-core BT11 represents the developing depth of the OS.

Cold centers: the local warmest isolated BT11 contour of the OS, which encloses only one cold core. The cold-center BT11 represents the warmest BT11 of it disconnecting from other OSs.

Segmentations: the OS outlines. The OS BT11 structure is simply characterized as the core and OS outlines. In the revised manuscript, "segmentations" has been replaced with "the OS outlines".

Anvil: the non-precipitating (precipitation less than 1 mm/hour) region in each OS.

(2) Terminology used in the target tracking:

Dynamic overlapping ratios (DORs): the OS is first moved to the locations predicted by the cross correlation and then overlaps with the OSs at the later moment. For the OS with single cores, three DOR indices are considered to determine the associations of two OSs at different times for whether they are the same object, including the DOR between cores, the DOR between OSs and the DOR between cores and OSs. In the revised manuscript, a cartoon illustration of the dynamic overlaps has been added as Fig. 1b.

Mergers and splits: Notably, the OS is not necessarily associated with only one OS. Mergers and splits are allowed and identified as the many-to-one and one-to-many OS associations, respectively. The mergers and splits in the fixed-BT11 and variable-BT11 tracking are very different. Mergers and splits in fixed-threshold tracking are dependent on the selection of the BT11 threshold. Owing to the selection of the fixed BT11 threshold, the identified targets are usually connected under a warmer threshold but are disconnected under a colder

threshold. As illustrated in Fig. 1f, if under the fixed threshold of 260 K, no mergers or splits occur. If under the fixed threshold of 220 K, the cutoff of the CCO by 220 K is the connected complex of multiple cores or two disconnected parts at different times. This change in the connecting conditions over time under the selected fixed threshold results in mergers and splits in fixed-threshold tracking. If under the fixed threshold of 200 K, the mergers and splits of cold cores are captured. It manifests that mergers and splits in fixed-threshold tracking can be attributed to many reasons: the threshold selection, the change in the connecting conditions and the variation in cold cores over time. In contrast, in variable-BT11 tracking, mergers and splits are not influenced by changes in the connecting conditions over time but is only related to the variation in cold cores as illustrated in Fig. 1e-f. In the revised manuscript, a cartoon illustration of Fig. 1e has been added to show the mergers and splits in the variable-BT11 tracking. Fig. 5 of the previous manuscript has been moved to Fig. 1f to illustrate the difference of the mergers and splits between the fixed-BT11 and variable-BT11 tracking.

**Table 1.** Summary of the key definitions for variable-$BT_{11}$ tracking developed in this study

| Name | Definition |
| --- | --- |
| Complex convective organizations (CCOs) | The contiguous area of the $BT_{11}$ colder than 260 K. |
| Organization segments (OSs) | The segmented single-core structural component of CCOs. |
| Cold-core $BT_{11}$ (OS developing depth) | The local coldest $BT_{11}$ contour in OSs. |
| Cold-center $BT_{11}$ (OS connecting depth) | The local warmest isolated $BT_{11}$ contour of only enclosing one core in OSs. |
| CCO $BT_{11}$ (CCO developing depth) | The coldest cold-core $BT_{11}$ of multiple cores in the CCO. |
| Anvil cloud | The non-precipitating (precipitation less than 1 mm/hour) region of each OS. |
| Dynamic overlapping rates (DORs) | The OS is moved to the location predicted by cross correlation and then overlaps with the OSs in the later image. |
| Merger and split $BT_{11}$ | The $BT_{11}$ of the merged cold core and the $BT_{11}$ of the splitting cold core. |
| Cold-core-peak $BT_{11}$ | The coldest cold-core $BT_{11}$ in lifecycles, representing the convective peaking strength. |
| Development and decay stages | The stage before and after the time of the cold core peaking at the coldest $BT_{11}$ (if there are multiple cores of the same $BT_{11}$, the one of the largest core areas is selected). |
| Lifecycle-accumulated duration, precipitation and anvil cloud amount | The accumulated time, precipitation and anvil cloud amount in the lifecycle. |

[Figure]

Figure 1. Illustrations of the variable-BT11 segment tracking algorithm. (a) Example illustrations of segmenting the CCO into single-core OSs as tracking targets. The CCO 3-dimensional structures in x, y and BT11 are identified by the adaptive variable-BT11 thresholds. The cold-core BT11 indicates the depth of development. The cold-center BT11 indicates the depth of the connection. (b) Example illustrations of tracking the OS by combining cross correlation and the overlap in areas. The OS is moved according to the displacement predicted by cross correlation and then overlaps with the OSs in the later images. (c-d) The dynamic overlapping situations of

two OSs of different moments when their cores have overlaps and no overlaps, respectively. The solid blue and green lines indicate the OS core and segmentation outlines of the current moment at the location predicted by cross correlation, respectively. The dashed blue and green lines indicate the OS core and segmentation outlines of the later moment, respectively. The gray cross indicates the non-association between OSs. (e) Examples illustrating tracked OS evolution (i.e., development, decay, mergers and splits). The red arrows indicate the evolution of the OS with time. (f) Illustrations of the difference between the variable-BT11 and fixed-BT11 tracking for mergers and splits. The solid red and blue lines are the CCO BT11 structures at different times captured by the adaptive variable-BT11 thresholds. The dashed red and blue contours are the mergers and splits captured by the fixed threshold of 220 K.

**2. Clarifying the methodology.** *The description of pattern-matching and the tracking algorithm were both quite confusing to me. The goals of each part of the analysis should be clearly laid out at the beginning of each section. It is confusing how segmentations, mergers, and splits are defined. I wish I could point to more specific aspects that I did not understand, but I am finding it difficult to do so at this point.*

**Response:** The description of pattern-matching and the tracking algorithm has been reorganized according to the steps of establishing the tracking algorithm with subtitles to indicate the goals of each step, as follows:

(1) Segmenting CCOs into the OSs of single cold cores;

(2) Tracking the displacement of OSs on the basis of cross correlation;

(3) Tracking OSs via dynamic overlaps;

(4) Quality control and validation of variable-BT11 segment tracking;

(5) Comparison with conventional fixed-threshold tracking.

Fig. 1 has been revised to help to understand the terminology definitions and steps of the tracking algorithm. In the revised Fig. 1, Fig. 1a illustrates the target identification by adaptive variable-BT11 thresholds. Fig. 1b illustrates the target tracking by the pattern-matching and dynamic overlaps; Fig. 1c-d illustrate the dynamic overlapping conditions. Fig. 1e-f illustrate the difference of the variable-BT11 tracking with the conventional fixed-BT11 tracking. Overall, Fig. 1 shows the flowchart of the tracking algorithm by cartoons.

The descriptions have also been revised according to the following minor comments. Please see more details in the responses to minor comments or in the revised manuscript.

Table 1 has been provided to summarize the terminology definitions for easily checking (see the response to the major comment #1). The mergers and splits have

been illustrated in the revised Fig. 1e. Mergers and splits are identified as the many-to-one and one-to-many OS associations, respectively.

The goal of the section 3 is to introduce the variable-BT11 tracking algorithm and its difference with the conventional fixed-BT11 tracking algorithm. The goal has been clarified at the beginning of section 3 as follows: "To distinguish the behaviors of clustered convective activities in CCOs, the organization segments (OSs) of single but variable-BT11 cold cores (Fig. 1a) are partitioned as tracking targets and are tracked by combining the cross correlation and the area overlap (Fig. 1b) based on the hourly infrared satellite images. This novel variable-BT11 segment tracking algorithm and its difference from the conventional fixed-threshold tracking algorithm are introduced in this section as follows.".

*3. Mergers & Splits statistics. Another thing to clarify is how statistics are computed for mergers and splits (e.g. Fig 7 and 8). How is a PDF of mergers and splits as a function of BT11 calculated? What if the two merging cores have different BT11? Which of the merging cores do the precip and anvil statistics represent? This was all very unclear.*

**Response:** In Fig. 7 and 8, the BT11 of mergers and splits refers to the merged cold-core BT11 and the splitting cold-core BT11, respectively, as illustrated in revised Fig. 1e. In this way, the BT11 of mergers and splits is represented by one splitting or merged core to compute the PDF as a function of BT11. The precipitation and anvil in Fig. 8 are the lifecycle-accumulated precipitation and anvil amount, which represent the accumulated precipitation and anvil of all OSs in a lifecycle. It has been clarified in the main text and the definitions have been summarized in Table 1.

*4. Cloud property results.*

- *I cannot find a description of how the anvil area is computed. Is it just the entire nonprecipitating area of each individual segment?*

  **Response:** Yes, the anvil is defined as the non-precipitating (precipitation less than 1 mm/hour) region of each OS.

  It has been clarified in the revised manuscript as "The OS can be further separated into precipitating and non-precipitating (precipitation less than 1 mm/hour) regions on the basis of the GPM. The non-precipitating area is identified as the anvil cloud. By segmentation, those precipitation and anvil pixels are explicitly associated with unique cold cores.".

  The definition can also be checked in Table 1.

- *The study region is (20S-20N, 90E-170E), which I find to be interesting from a cloud property perspective. I imagine this choice was largely motivated the availability of different observations, especially the ARM sites. The region*

*includes some of the western Indian Ocean and Bay of Bengal, the entire Maritime Continent region, and some of the west Pacific warm pool. The characteristics of convective systems can differ significantly between the maritime continent, where land -based convection dominates, and the oceanic regions, where larger mesoscale convective systems are typical (see Fig 9 in Yuan & Houze 2010, doi:10.1175/2010JCLI3671.1). Does it make sense to aggregate the precipitation, duration, and anvil area statistics across this entire region? I imagine there would be considerable differences between the Maritime Continent and the oceanic regions, with smaller cloud systems and fewer mergers/splits for land-based convection. The author could consider stratifying the results by region, or at the very least acknowledging what I imagine are very large spreads within each BT11 bin for the cloud property statistics.*

**Response:** In consideration of the significant difference of the convective systems between the maritime continent and oceanic regions, we only focus on the oceanic regions in Section 4 in the revised manuscript. The warm pool of the tropical western Pacific Ocean (130°W-170°E, 20°S-20°N) is a typical region of the oceanic convection precipitating and producing anvil clouds (Wall et al., 2018). In Section 4, only the OS lifecycles over oceans of this region are considered for investigating the behaviors of the oceanic convection precipitating and producing anvil clouds. This has been clarified at the beginning of Section 4.

***5. Grammar and Structure.*** *As a native English speaker, I found this paper quite difficult to understand at times, and this is likely a major reason for the perceived lack of clarity. I simply want to share that thought with the author, so that they can adjust and edit as they see fit. If editing services are available at the author's institution, they may wish to pursue them. This is simply a suggestion, and I do not consider it necessary for the paper to be published, as long as the necessary components are greatly clarified.*

**Response:** Thanks. To improve the readability, the revised manuscript has been better structured by adding more subtitles for showing the goals of each analysis. The descriptions have been adjusted and reedited with more examples and details to improve the clarity according to comments from referees. Editing services have been used to help to correct grammar mistakes and to revise the unclear descriptions. Careful proofreading has been done by authors.

***More Minor Comments:***

- *What is meant by "organization segments"…does this just mean the different structural components of the storm?*

**Response:** Yes, the OS is the segmented single-core structural component of the complex organizations. The OS is the tracking target of the variable-BT11 tracking algorithm.

It has been clarified in the revised manuscript as "As illustrated in Fig. 1a, the CCO is the complex organization of multiple connected convections and is identified as the contiguous area of the BT11 that is colder than 260 K. The 260-K threshold can enclose 95% of deep convective clouds and as much of the anvil cloud as possible but with the least contamination from lower-level clouds (Yuan and Houze, 2010; Yuan et al., 2011; Chen and Houze, 1997). The segmented single-core structural component of CCOs is identified as the OS to be used as the tracking target."

A cartoon in Fig. 1a might be helpful to understand the OS definition.

- *Line 30: "due to **the fact** that the…"*

**Response:** It has been corrected.

- *Line 43-46: the author cites three papers as evidence that convective organization and precipitation efficiency (PE) are related, but I am not sure these references are correct. Bao & Sherwood (2019), https://doi.org/10.1002/2018MS001503, seems like a more appropriate reference here. Choi et al (2017) found that greater PE (by their definition of PE) was associated with reduced cirrus cloud area, but this is not the same thing as convective organization. Lindzen et al (2001) and Mauritsen & Stevens (2015) hypothesized about the relationship between PE and anvil cloud area, but did not present any evidence of a relationship between organization and PE.*

**Response:** It has been corrected.

- *Line 51: what are the two distinct modes of convection being referred to here?*

**Response:** The two distinct modes are in terms of the BT11 and refer to the deep convective clouds and anvil clouds.

This sentence has been corrected as: "From the images of the brightness temperature at 10.8 μm (BT11) of geostationary satellites (GEOs), pixels of thin cirrus clouds cannot be accurately distinguished from cloudless pixels, but the major structure of the organized convection, consisting of the deep convective clouds and the associated anvil clouds, can be observed continuously in time and used for tracking".

- *Line 81: replace "190 W" with "170 E"*

**Response:** It has been corrected.

- *Line 128: the equation for the speed bias (eq 1) is incorrect. The subscripts are switched around. See eq 4 in Nieman et al (1997)*

**Response:** It has been corrected.

- *Line 128: latter -> later*

**Response:** It has been corrected.

- *Section 2.5: this section was very unclear to me. Please provide some context for what the goal of this pattern matching is and how it fits in to the tracking algorithm*

**Response:** The goal of the pattern matching based on the cross correlation is to predict the displacement of OSs. The OS is moved to the location predicted by the cross correlation and then overlaps with the OSs in the later image. In this way, the dynamic overlaps by combining the cross correlation and the overlap in areas are used for tracking the OS, which avoids the mistakes in tracking the fast-moving OS.

To better introduce the goal of the pattern matching and how it fits the tracking algorithm, this subsection has been rephrased and is reorganized into the Section 3 of the tracking algorithm in the revised manuscript. And a cartoon subfigure has been added as Fig. 1b for explaining how the pattern matching is achieved and how it fits the tracking algorithm. The pattern matching is a necessary step to find the OS movement for accomplishing the dynamic overlap.

  - *Line 142: "normalized BT11"…normalized in what sense?*

    **Response:** The BT11 in the target scene and the BT11 in each cross scene are normalized, respectively. The patten-matching displacement is determined by the minimum of the sum of squared differences (SSD) of the normalized BT11 between the OS target scene and the cross scene.

    This has been clarified in the revised manuscript as: "The BT11 pattern of the target scene is normalized, and so is the BT11 pattern of each cross scene. The patten-matching displacement is determined by the minimum of the sum of squared differences (SSD)

of the normalized BT11 between the OS target scene and the cross scene".

o  *Define "target scene" and "cross scene"*

**Response:** The definitions of the target and cross scenes have been added in the main text: "the target scene is the OS BT11 pattern. The search region is centered at the core centroid of the target and confined to a radius of 50 km, which corresponds to a maximum OS motion of 50 km/hour (Merrill et al., 1991). The cross scene has the same shape as the OS target and refers to all possible scenes to match the OS target within the search region.".

Fig. 1b has been added for illustrating these definitions.

o  *Line 145-146: "for the areas larger…" what areas are you talking about?*

**Response:** It is the area of the tracking target. The organization segment of irregular shapes is selected as the tracking target. For large (small) targets, a lower (higher) threshold for the pattern correlation is used to examine the pattern matching. It has been clarified in the manuscript.

o  *It seems that SSD would be minimized if the BT11 field does not change at all from one time to the next. Are the fields adjusted in space to overlap? Is this what normalization refers to? This was generally quite confusing.*

**Response:** Yes, if the BT11 field does not change at all, the SSD is minimized when the displacement is zero. The BT11 fields are not adjusted to overlap, but the location of the target is adjusted according to the predicted displacement and then to overlap. The normalization refers to that the BT11 in the target scene and the BT11 in each cross scene are normalized, respectively, then are matched to calculate the SSD.

A cartoon figure has been added as Fig. 1b, which could be helpful to understand the cross correlation. The target scene refers to the OS with irregular shapes. The cross scene has the same shape to the target scene, and is within the search region of the radius 50 km. The BT11 in the target scene and the BT11 in the cross scene are normalized, respectively, and then are matched to calculate the SSD. Then, the location of the target is adjusted according to the

displacement predicted by cross correlation and then overlaps with the OSs in the later image.

The flowchart is as follows:

(1) Segmenting the complex organizations into the organization segment (OS) of single cores (Fig. 1a);

(2) Tracking the displacement of the OS based on the cross correlation (Fig. 1b);

(3) Moving the OS to the locations predicted by the cross correlation and then computing the overlaps with the OSs at the later moment (referring to dynamic overlaps) (Fig. 1b).

(4) Tracking the OS by the dynamic overlaps (Fig. 1c-d).

For clarity, the main text has been reorganized according to the flowchart and each part has been added a small title to clarify the purpose.

- *Fig 1:*

  o *the font size in panel 1a is too small at the top of the figure ("centers" and "cores"). The green font color for "connecting depth" and "Developing depth" is hardly visible.*

  **Response:** The font size of Fig. 1a has been enlarged. The font color has been modified.

  o *What does "after moving" mean in the legend? Aren't you showing two moments in time, with the dotted lines indicating the later moment? Aren't the solid lines then showing the "before moving" picture?*

  **Response:** "after moving" means that the tracking target is moved to the predicted location by the cross correlation (the solid lines) and then overlaps with the targets in the later images (the dash lines). Yes, the solid and dash lines indicate the targets at the current and later moments, respectively. But, by the cross correlation, the displacement of the current target is predicted and the target is first moved to the predicted location and then overlaps with the targets in the later images.

  A more detailed illustration of dynamic overlaps has been added as Fig. 1b.

> ○ *Panels b,c: do the displacements between the solid and dotted lines reflect displacement over time? Or have the later moments been pattern-matched and adjusted for maximum overlap?*

**Response:** The displacement between the solid and dotted lines does not reflect the displacement over time. The location of the target at the current moment has been adjusted by the cross correlation before overlapping with the targets at the later images. Fig. 1b has been newly added and could be helpful to explain how the dynamic overlaps are computed. As shown in Fig. 1b, the dynamic overlaps refer to the overlap between the cross scene of the min SSD and the target at the later image.

- *Line 180: If I am understanding correctly, the algorithm detects the full cloud segment by expanding out from the core in 1K BT11 intervals. I imagine there is some ambiguity at times, in which it is not obvious which core a piece of anvil cloud should be assigned to? How is this dealth with?*

**Response:** For segmentation, the pixels lying outside the centers are assigned to the connected neighborhood OSs by the 1-K interval. To be specific, all BT11 contours of the 1-K interval between the cold-center BT11 and 260 K need to be found first. The assignment of the pixels outside the centers is conducted in the order from cold to warm BT11 contours of the 1-K interval. The initial OS is just the center and it is updated after every 1-K-interval assignment. An example illustration of the 1-K-interval assignment is shown in Fig. 2. On the basis of the 8-point-connected neighborhood in which the 8 surrounding points are recognized as the connected neighborhood to the center point, the distance between two pixels is computed as the number of necessary pixels connecting them. According to the nearest linear distance, as shown in Fig. 2a, some of the pixels assigned to OS2 (those light green pixels in Fig. 2a) are disconnected from OS2 but connected to OS1. After the assignment, OS2 is composed of two disconnected parts. For an organized convective system, the assigned pixels outside the center can also be understood as outflowing anvil clouds from the center. It would be strange that the outflowing anvil clouds from OS2 are not connected with its original OS2 but connected with OS1. To avoid these conditions, the distance of the nearest route is used to determine the pixel assignment. Here, the route of OS1 and OS2 to reach a pixel (the blue and red arrows in Fig. 2b) is confined to within the 1-K-interval contour. Pixels of the same distance to OS1 and OS2 are randomly assigned. In Fig. 2b, the assignment of the pixels on the basis of the distance of the nearest route is more reasonable than that in Fig. 2a on the basis of the nearest linear distance. Thus, in every 1-K-interval assignment, the distance of the nearest route is used to accomplish the segmentation and the OSs are updated with these newly assigned pixels iteratively until all the pixels within the CCO are assigned.

This has been clarified in the revised manuscript.

[Figure]

Figure 2. Illustrations of segmentation according to the nearest linear distance (a) and the nearest route distance (b). The dark blue and green pixels represent the OS1 and OS2 centers, respectively. The colored pixels outside the centers are the pixels to be assigned in the contour of the cold-center BT11 plus 1 K. The light blue and green pixels are assigned to OS1 and OS2, respectively. The numbers inside those pixels indicate the number of necessary pixels to connect with OS1 and OS2, respectively. The arrows in (a) and (b) represent the nearest distances of OS1 and OS2 to reach the yellow-edge pixel, as examples to illustrate the computations of the linear distance and the route distance, respectively.

- *Lines 185-190: this paragraph was quite confusing to read, and I had to read it about 5 times to understand the details here. Cold-center BT11, complex BT11, and cold-core BT11 should be more clearly defined somewhere…at the moment they are buried in Fig 1a.*

**Response:** Table 1 has been added to provide a summary of the key definitions in this study.

- *Line 200: for clarity, specify "The core-core and segmentation-segmentation DORs are relative to the minimum area… The core-segmentation DORs are relative to the core area"*

**Response:** These definitions have been clarified in the main text as "For the OS with the core structure, three indices of the dynamic overlapping ratio (DOR) are considered to determine the associations of two OSs at different times for the same object, including the DOR between cores, the DOR between OSs and the DOR between cores and OSs. The DOR between cores is the ratio of their overlaps in cores relative to the minimum area of the cores to represent the degree of core overlap. The DOR between OSs is the ratio of the OS overlap relative to the minimum area of the OS to represent the degree of OS overlap. The DOR between cores and OSs is the ratio of the overlap of the OS to the core relative to the core area, representing the degree of the core overlapped by the later or previous OS.".

• *Lines 198-205 – this paragraph is also confusing to read. Does "temporal associations" mean that the you consider it to be the same storm at different times?*

**Response:** Sorry about that. Yes, the temporal associations indicate the same storm at different times. It has been explained in the main text.

This paragraph has been rephrased as "Two OSs of different moments are associated in time and considered the same object when these two OSs overlap sufficiently. The overlapping situations of two OSs are distinguished by whether their cores overlap with each other (Fig. 1c) or not (Fig. 1d). Those pairs of OSs in situations (ⅰ), (ⅱ) and (ⅳ) in Fig. 1b all sufficiently overlap with the DOR between either cores or OSs greater than 50% and thus are associated in time to reflect the OS evolution with time. The situation (ⅲ) in Fig. 1c with DORs of both cores and OSs less than 50% indicates that these two OSs have no associations. In Fig. 1d, when the cores of two OSs do not overlap, the determinant of the OS association relies on the DOR between OSs and the DOR of OSs to cores. In those cases, the OSs are associated in time only in situation (ⅱ) in Fig. 1d, with those two DOR indices both larger than 50%. Those pairs of OSs in the other situations in Fig. 1d are obviously not associated. Overall, if the DORs of two OSs satisfy the overlapping conditions of (ⅰ), (ⅱ) and (ⅳ) in Fig. 1c and (ⅱ) in Fig. 1d, they are associated in time.".

• *Fig 2 middle row: the arrows were a bit confusing, maybe it could be equally effective to just put red and white dots on each panel (optional suggestion).*

**Response:** The arrows have been replaced with dots.

• *Line 241: unclear: "thus ends by less disconnected convection complex". It looks less connected, not less disconnected.*

**Response:** It has been corrected as "The major branch (the red line in the middle panel of Fig. 5) begins with the large complex organization of connected convections but ends with only one of disconnected parts."

• *Line 244: "evolution of the system structures but not the variations of the connections?"…what does this mean?*

**Response:** Mergers and splits in fixed-threshold tracking are dependent on the selection of the BT11 threshold. Owing to the selection of the fixed BT11 threshold, the identified targets are usually connected under a warmer threshold but are disconnected under a colder threshold. As illustrated in Fig. 1f, if under the fixed threshold of 260 K, no mergers or splits occur. If under the fixed threshold of 220 K, the cutoff of the CCO by 220 K is the connected complex of multiple cores or two disconnected parts at different times. This

change in the connecting conditions over time under the selected fixed threshold results in mergers and splits in fixed-threshold tracking. If under the fixed threshold of 200 K, the mergers and splits of cold cores are captured. It manifests that mergers and splits in fixed-threshold tracking can be attributed to many reasons: the threshold selection, the change in the connecting conditions and the variation in cold cores over time. In contrast, in variable-BT11 tracking, mergers and splits are not influenced by changes in the connecting conditions over time but is only related to the variation in cold cores as illustrated in Fig. 1e-f.

Examples are shown in Fig. 5. In the fixed-threshold tracking of 210K (the middle panel of Fig. 5), the mergers and splits are caused by the variations of whether convections are connected or not under the 210-K threshold. In the variable-BT11 tracking (the bottom panel of Fig. 5), the tracked mergers and splits are the mergers and splits of cold cores and are not influenced by the variations in the connecting conditions with time. This explanation has been added to the revised manuscript.

- *Figure 5 is completely lost on me – I do not know what this figure is trying to show, and the caption is not very helpful here. Please explain this figure.*

**Response:** Figure 5 has been modified to be the Fig. 1f in the revised manuscript. The caption has been revised as "Illustrations of the difference between the variable-BT11 and fixed-BT11 tracking for mergers and splits. The solid red and blue lines are the CCO BT11 structures at different times captured by the adaptive variable-BT11 thresholds. The dashed red and blue contours are the mergers and splits captured by the fixed threshold of 220 K."

In the main text, it has been explained as "mergers and splits in fixed-threshold tracking are dependent on the selection of the BT11 threshold. Owing to the selection of the fixed BT11 threshold, the identified targets are usually connected under a warmer threshold but are disconnected under a colder threshold. As illustrated in Fig. 1f, if under the fixed threshold of 260 K, no mergers or splits occur. If under the fixed threshold of 220 K, the cutoff of the CCO by 220 K is the connected complex of multiple cores or two disconnected parts at different times. This change in the connecting conditions over time under the selected fixed threshold results in mergers and splits in fixed-threshold tracking. If under the fixed threshold of 200 K, the mergers and splits of cold cores are captured. It manifests that mergers and splits in fixed-threshold tracking can be attributed to many reasons: the threshold selection, the change in the connecting conditions and the variation in cold cores over time. In contrast, in variable-BT11 tracking, mergers and splits are not influenced by changes in the connecting conditions over time but is only related to the variation in cold cores as illustrated in Fig. 1e-f."

- *Figure 6*

  o *It would be nice to add a panel showing the sample size for each cold-core-peak BT11 bin.*

  **Response:** Figure 6 has been added in the revised manuscript to show the sample size for each cold-core-peak BT11 bin, as shown below.

[Figure]

Figure 6. Sample numbers of tracked OS lifecycles with cold-core-peak BT11 values from 185-255 K in the tropical western Pacific (130°W-170°E, 20°S-20°N) in 2006. The contribution fraction of the OS lifecycles to the precipitation and anvil cloud amount is shown on the right axis.

  o *it would be nice to see the spreads in duration, precip, and anvil amount for each cold-core-peak BT11 bin. The t-test for the mean is nice, but I imagine these is a very large spread on these quantities, since convective systems vary greatly in size. It would be good to show the spread if there is a simple way to do so.*

  **Response:** The PDFs of the lifecycle-accumulated duration, precipitation and anvil amount for each cold-core-peak BT11 have been added in Figure. 7a-c in the revised manuscript, as shown below.

[Figure]

Figure 7. PDFs of the accumulated duration (a), precipitation (b) and non-precipitating anvil amount (c) of the OS lifecycles of different cold-core-peak BT11 values from 190-250 K. The mean accumulated duration (d), precipitation (e) and non-precipitating anvil amount (f) contributed by the development (blue lines) and decay stages (red lines) as a function of the cold-core-peak BT11 from 185-255 K. The black lines represent the differences in the accumulated duration, precipitation and anvil between the development and decay stages in (d-f), respectively. The error bars indicate the 95% confidence intervals of the means based on the t test.

- *Line 269-270: "and the difference of the duration, precip, and anvils between two stages has exponential increases with the core peaking at colder BT11." I am struggling to see how this is the case in Fig 6. It does not seem like the difference between the orange and blue lines is exponentially greater at lower BT11, although it is hard to tell because of the log scale.*

**Response:** The difference of the duration, precipitation and anvils between two stages has been shown in Figs. 7d-f by the solid black lines. It can be seen the difference roughly has an exponentially increase with the core peaking at colder BT11.

- *Lines 278-281: this sentence is not clear, please revise. Will be helpful once HCS is clearly defined. For example, how do mergers and splits create more HCS? My initial thought was that HCS referred to the entire system BT11<260, including many segmentations?*

**Response:** The HCS in the previous manuscript is just the segmented single-core structural components. To avoid misunderstandings, we directly use the

"organization segments (OSs)" to replace the "HCS". Here, we mean that the mergers and splits would create more OSs in the lifecycle to increase the precipitation and anvils.

- *Equations 7 & 8: what is N exactly? The definition of HCS seems to be very important here. Is it the number of segments? Also, it would be helpful to explain what the point of this sort of analysis is before showing the results.*

**Response:** The N is the lifecycle-accumulated number of OSs. Yes, it is the number of segments. It has been clarified in the revised manuscript as: "N is the accumulated OS number in the lifecycle".

The purpose of this analysis has been clarified at the beginning before discussing the results as: "How do mergers and splits influence the lifecycle-accumulated precipitation and anvil cloud amounts? There are two possible mechanisms: the hourly precipitation and anvil production of each OS in the lifecycle are enhanced, and the accumulated number (N) of OSs in the lifecycle is increased.".

---

## Author Comment (AC2)

**Response to Anonymous Referee #2**

*Referee #2: This manuscript describes a method to track convective systems in the tropics using a so-called 'variable-BT' method. The tracked objectives are evaluated against observations by comparing object drifting speed and direction with those observed from three ARM sites. Contributions of convective activities to precipitation and anvil amount are discussed. I think the topic can be a good contribution to the community by bringing a more flexible tracking framework. However, I believe this manuscript needs substantial improvements before it can be considered for publication.*

**Response:** We thank anonymous referee for reviewing our manuscript and very helpful comments to modify the manuscript. We have responded to all comments and carefully improved the representation of the manuscript accordingly.

*Major:*

1. ***Grammar and Readability:***

*There are numerous grammar errors that make the manuscript difficult to read. The author should do a thorough proof-reading or seek help from a professional editing service before submitting the revised manuscript. Particular attention should be paid to the abstract, as a readable abstract is more likely to attract readers' interest in the method developed and can help increase the paper's impact.*

**Response:** Professional editing service has been used to correct the grammar mistakes and unclear descriptions. Careful proofreading has been done by authors. To improve the readability, the revised manuscript has been reorganized with more subtitles for showing the goal of analyses.

The abstract has been reorganized into two paragraphs, to introduce the innovation of the tracking algorithm; and the convective processes revealed by the novel tracking algorithm. It has been reedited as: "The convective processes of precipitation and the production of anvil clouds determine the Earth's water and radiative budgets. However, convection could have very complicated convective organizations and behaviors in the tropics. Many convective activities in various life stages are connected in complex convective organizations, and it is difficult to distinguish their behaviors. In this work, on the basis of hourly infrared brightness temperature (BT) satellite images, with a novel variable-BT tracking algorithm, complex convective organizations are partitioned into organization segments of single cold cores as tracking targets. The detailed evolution of the organization structures (e.g., the variation in the cold-core BT, mergers and splits of cold cores) can be tracked, and precipitation and anvil clouds are explicitly associated with unique cold cores. Compared with previous tracking algorithms that focused only on variations in areas, the novel variable-BT tracking algorithm is capable of documenting the evolution of both the area and BT structures. For validation, the

tracked motions are compared against the radiosonde cloud-top winds, with mean speed differences of -1.6 m/s and mean angle differences of 0.5°.

With the novel variable-BT tracking algorithm, the behaviors of oceanic convection over the tropical western Pacific Ocean are investigated. The results show that the duration, precipitation and anvil amount of the lifecycle accumulation all have simple loglinear relationships with the cold-core-peak BT. The organization segments of the peak BT values less than 220 K are long-lived, with average durations of 4-16 hours, whereas the organization segments of the warmer-peak BT values disappear rapidly within a few hours but with a high occurrence frequency. The decay process after the cold core peaks contributes to more precipitation and anvil clouds than does the development process. With the core peaking at a colder BT, the differences in the accumulated duration, precipitation and anvil production between the development and decay stages increase exponentially. Additionally, the occurrence frequency of mergers and splits also has a loglinear relationship with the cold-core-peak BT. For the lifecycles of the same cold-core-peak BT, the lifetime-accumulated precipitation and anvil amount are strongly enhanced in complicated lifecycles with the occurrence of mergers and splits compared with those with no mergers or splits. For the total tropical convective cloud water budget, long-lived complicated lifecycles make the largest contribution to precipitation, whereas long-lived complicated and short-lived simple lifecycles make comparable contributions to the anvil cloud amount and are both important."

**2. *Introduction:**

*The author spent most of the space describing the importance of segmenting convective systems, but the motivation for the work in this manuscript is not well articulated. While there are already quite several tracking algorithms in the community, why is the tracking method developed here a necessary contribution? What are the major differences/advantages of your tracking method over others? Why is it important to have the extra features (if any) from your tracking algorithm? This information should be added to either the introduction or the discussion.*

**Response:** A paragraph has been added to introduce the motivation of this work and the advantages of the tracking algorithm developed in this work, as follows: "In this work, complex convective organizations (CCOs) are segmented into simple structural components of single cold cores and tracked separately according to variable-BT11 identification and dynamic overlap. Compared with fixed-threshold tracking, the variable-BT11 tracking algorithm has the advantages of documenting more detailed convective evolution in CCOs. Although several variable-BT11 tracking algorithms have been proposed, the tracked lifecycle is still described mostly by the variation in areas and lacks of the BT11 structural information. By the novel variable-BT11 tracking algorithm developed in this work, the tracked lifecycle is described by the cold-core BT11 variation in the CCO structural components. The precipitation and non-precipitating anvil clouds are explicitly associated with unique cold cores."

**3. Flow and Logic:**

*The flow and logic of the manuscript need improvement. For example, the paragraph starting from L245 and Figure 5 should be moved up to before Figure 3 or even earlier. The L245 paragraph introduces one of the key novelties of the method developed in this manuscript compared to previous fixed-BT tracking methods, and thus should be introduced and highlighted earlier before demonstrating and evaluating the results in Figure 4 and Figure 3, respectively.*

**Response:** The flow and logic of the Section 3 has been reorganized. Figure 5 has been moved to be the subfigure (f) in the revised Fig. 1. And the L245 paragraph has been introduced and highlighted earlier before the discussing the results of Fig. 3 and 4.

Overall, for readability, subtitles in section 3 have been added to help grasp the goal of analyses and the step of establishing the tracking algorithm, as follows:

(1) Segmenting CCOs into the OSs of single cold cores;

(2) Tracking the displacement of OSs on the basis of cross correlation;

(3) Tracking OSs via dynamic overlaps;

(4) Quality control and validation of variable-BT11 segment tracking;

(5) Comparison with conventional fixed-threshold tracking.

The key novelties of the tracking method developed in this manuscript in comparison to the fixed-threshold tracking are highlighted at the start of the subsection (5) as: "The fundamental difference between fixed-threshold and variable-BT11 tracking is target selection. With the fixed threshold of the BT11, the connected convection of multiple cold cores is recognized as tracking targets, and only the area information is accessible. With the OS as tracking targets, variable-BT11 tracking is capable of documenting the detailed evolution of each OS within CCOs, such as the developing depth, connecting conditions, and contributions to precipitation and anvil clouds.".

**4. Limitations in ARM observations**

*MMCR is a millimeter wavelength radar, and the signal attenuates quickly when observing deep convective clouds, especially in convective core and stratiform regions. The cloud top heights from MMCR in these regions are thus underestimated if relying on ARSCL data for detection. Cloud fraction profiles are also significantly impacted in the upper part of the convective systems. This will likely contribute to the discrepancies in the comparison between the HCS-drift winds and radiosonde cloud-top winds in Figure 3.*

**Response:** Thanks. This limitation due to the beam attenuation has been clarified in the clarified in the main text: "some bias might be attributed to the uncertainty in the cloud-top heights. For its detection, the MMCR might underestimate the cloud

top height since its signal would attenuate quickly for deep convective clouds (Hollars et al., 2004). In the convective systems, the motion of air is highly organized (Houze, 2004); thus, system movement might be inconsistent with the observed winds at the cloud-top height.".

*Minor:*

*Is BT the only parameter used in identifying segments? How did you segment the objects from the BT thresholds? Was it a watershed-type segmentation? The author does not demonstrate well how the 'variable-BT11' method works, with details lacking and thus making it hard to evaluate the method's appropriateness.*

**Response:** Yes, the BT is the only parameter used in identifying segments. Figure 2 in the revised manuscript (as shown below) has been added to illustrate how to segment the objects.

The details of the segmentation have been clarified as in the revised manuscript as: "For segmentation, the pixels lying outside the centers are assigned to the connected neighborhood OSs by the 1-K interval. To be specific, all BT11 contours of the 1-K interval between the cold-center BT11 and 260 K need to be found first. The assignment of the pixels outside the centers is conducted in the order from cold to warm BT11 contours of the 1-K interval. The initial OS is just the center and it is updated after every 1-K-interval assignment. An example illustration of the 1-K-interval assignment is shown in Fig. 2. On the basis of the 8-point-connected neighborhood in which the 8 surrounding points are recognized as the connected neighborhood to the center point, the distance between two pixels is computed as the number of necessary pixels connecting them. According to the nearest linear distance, as shown in Fig. 2a, some of the pixels assigned to OS2 (those light green pixels in Fig. 2a) are disconnected from OS2 but connected to OS1. After the assignment, OS2 is composed of two disconnected parts. For an organized convective system, the assigned pixels outside the center can also be understood as outflowing anvil clouds from the center. It would be strange that the outflowing anvil clouds from OS2 are not connected with its original OS2 but connected with OS1. To avoid these conditions, the distance of the nearest route is used to determine the pixel assignment. Here, the route of OS1 and OS2 to reach a pixel (the blue and red arrows in Fig. 2b) is confined to within the 1-K-interval contour. Pixels of the same distance to OS1 and OS2 are randomly assigned. In Fig. 2b, the assignment of the pixels on the basis of the distance of the nearest route is more reasonable than that in Fig. 2a on the basis of the nearest linear distance. Thus, in every 1-K-interval assignment, the distance of the nearest route is used to accomplish the segmentation and the OSs are updated with these newly assigned pixels iteratively until all the pixels within the CCO are assigned.".

[Figure]

Figure 2. Illustrations of segmentation according to the nearest linear distance (a) and the nearest route distance (b). The dark blue and green pixels represent the OS1 and OS2 centers, respectively. The colored pixels outside the centers are the pixels to be assigned in the contour of the cold-center BT11 plus 1 K. The light blue and green pixels are assigned to OS1 and OS2, respectively. The numbers inside those pixels indicate the number of necessary pixels to connect with OS1 and OS2, respectively. The arrows in (a) and (b) represent the nearest distances of OS1 and OS2 to reach the yellow-edge pixel, as examples to illustrate the computations of the linear distance and the route distance, respectively.

*The 'feature-matching displacement' section 2.5, how is this matrix used in the method?*

**Response:** The section 2.5 has been reorganized into the section 3 and a cartoon subfigure has been added into Fig. 1 to illustrate how the feature-matching displacement is used (as shown below).

[Figure]

Figure. 1b in the revised manuscript. (b) Example illustrations of tracking the OS by combining cross correlation and the overlap in areas. The OS is moved according to

the displacement predicted by cross correlation and then overlaps with the OSs in the later images.

In the main text, it has been clarified as: "In Fig. 1b, to track the temporal evolution of OSs, the OS is moved to the location predicted by cross correlation and then overlaps with the OSs in the later image. In this way, the dynamic overlaps can be used to tolerate the fast-moving OS in tracking.".

*What is the minimum temporal resolution required to perform the tracking? You mentioned in Section 2.1 the 1-hour resolution BT images from CERES. Are those the images you used for tracking?*

**Response:** The minimum temporal resolution is 1 hour. Yes, the 1-hour GEO BT11 images from the CERES project are used for tracking. If the missing time gap between two continuous images exceeds 2 hours, the OSs in these two images and all lifecycles including these OSs are excluded from analyses. Additionally, the OS touching the image edges and all lifecycle including the OS touching edges are excluded. This has been clarified in the revised manuscript.

***Specific:***

*L80: GEO should be defined.*

**Response:** It has been defined at the line 50 in the revised manuscript: "geostationary satellites (GEOs)".

*L83: Only data from the year 2006 was used?*

**Response:** Yes, only data from the year 2006 was used in this work.

*L130: The symbols in the formulas should be explained.*

**Response:** It has been explained as "Here, the mean speed and angle bias, the mean vector difference (MVD), the standard deviation (SD) of the MVD and the root-mean-square error (RMSE) of the tracked cloud motions compared with the observational cloud-top winds were computed. U and V are the x- and y-component winds, respectively. The subscripts i and r indicate an individual sample of the tracked cloud motion and the corresponding reference cloud-top winds of radiosondes, respectively, and N is the total number of samples.".

*L138-140: This sentence is unclear. Please rephrase it. When you say 'irregular segments', how do you determine the irregularity? What about segments with relatively regular shapes like convective core regions?*

**Response:** The newly added Fig. 1b (as shown above) in the revised manuscript might be helpful to explain the irregularity. Here, the irregularity means the target scene in the cross correlation is not the regular square box but the segmented organization components with irregular shapes.

This sentence has been rephrased as: "As shown in Fig. 1b, the target scene is the OS BT11 pattern. The search region is centered at the core centroid of the target and confined to a radius of 50 km, which corresponds to a maximum OS motion of 50 km/hour (Merrill et al., 1991). The cross scene has the same shape as the OS target and refers to all possible scenes to match the OS target within the search region. The BT11 pattern of the target scene is normalized, and so is the BT11 pattern of each cross scene. The patten-matching displacement is determined by the minimum of the sum of squared differences (SSD) of the normalized BT11 between the OS target scene and the cross scene.".

*Figure 2: How was this figure generated? Is it from hypothetical data or satellite observations? How many years of data are used? Can you add the sample number to the figure?*

**Response:** The cloud-top winds are derived by combing the radar and radiosonde observations at those sites (see more details in Sect. 2.3) as the observational reference to examine the tracked OS motions from the hourly satellite images in 2006. To collocate the observations from the ground-based sites and satellites, the tracked OS-drift winds from the GEO observations that are closest to the time of the cloud-top wind observations and nearest to the site locations are used to compare with the cloud-top winds at those ground-based sites. The observational time difference is no more than one hour and the tracked OS core centroid is within 150 km of those ARM site locations. These are consistent with the previous studies for examining the performance of cloud-drift winds (Nieman et al., 1997; Santek et al., 2019; Daniels et al., 2020). This has been clarified in the revised manuscript.

It is from the satellite observations and not from the hypothetical data. One-year data in 2006 is used. The sample number has been added in the top left-hand corner.

*L184: What is the difference between cold-core BT and cold-center BT? The previous paragraph does not seem to describe the terminology well.*

**Response:** Figure 1a has been revised to better illustrate the terminology definitions and Table 1 has been added to summarize these definitions for easily checking (as shown below).

[Figure]

Figure. 1a in the revised manuscript. (a) Example illustrations of segmenting the CCO into single-core OSs as tracking targets. The CCO 3-dimensional structures in x, y and BT11 are identified by the adaptive variable-BT11 thresholds. The cold-core BT11 indicates the depth of development. The cold-center BT11 indicates the depth of the connection.

The innermost ring of the local coldest BT11 is defined as the cold core for the most active vertically developing region in the OS. The BT11 of the cold core represents the OS developing depth. The isolated ring of the warmest BT11 is the cold center and the OS would be connected (disconnected) to the surrounding OSs outside (within) the center. Thus, the cold-center BT11 can be used to indicate the connecting condition between the OSs in the CCO. The complexity of convective organizations can be inferred from the cold-center BT11 of OSs. Only when the cold-center BT11 is 260K, the OS is the isolated convective body and disconnected with other OSs. These descriptions have been added in the main text.

**Table 1.** Summary of the key definitions for variable-$BT_{11}$ tracking developed in this study

| Name | Definition |
| --- | --- |
| Complex convective organizations (CCOs) | The contiguous area of the $BT_{11}$ colder than 260 K. |
| Organization segments (OSs) | The segmented single-core structural component of CCOs. |
| Cold-core $BT_{11}$ (OS developing depth) | The local coldest $BT_{11}$ contour in OSs. |
| Cold-center $BT_{11}$ (OS connecting depth) | The local warmest isolated $BT_{11}$ contour of only enclosing one core in OSs. |
| CCO $BT_{11}$ (CCO developing depth) | The coldest cold-core $BT_{11}$ of multiple cores in the CCO. |
| Anvil cloud | The non-precipitating (precipitation less than 1 mm/hour) region of each OS. |
| Dynamic overlapping rates (DORs) | The OS is moved to the location predicted by cross correlation and then overlaps with the OSs in the later image. |
| Merger and split $BT_{11}$ | The $BT_{11}$ of the merged cold core and the $BT_{11}$ of the splitting cold core. |
| Cold-core-peak $BT_{11}$ | The coldest cold-core $BT_{11}$ in lifecycles, representing the convective peaking strength. |
| Development and decay stages | The stage before and after the time of the cold core peaking at the coldest $BT_{11}$ (if there are multiple cores of the same $BT_{11}$, the one of the largest core areas is selected). |
| Lifecycle-accumulated duration, precipitation and anvil cloud amount | The accumulated time, precipitation and anvil cloud amount in the lifecycle. |

*Figure 6: Did you explain how you define the development and dissipation stages somewhere? Are the results shown in Figure 6 (and subsequent figures) from above the three ARM sites, or from the tropics in general as specified at the beginning of Section 2.1? Can you add sample numbers to either the figure or the caption?*

**Response:** The development (decay) stage is defined as the stage before (after) the time of the cold core peaking at the coldest BT11 with the largest core area. It has been clarified in the manuscript and the definition can be checked in Table 1.

The results in Section 4 from Figs. 6-10 are all from the tropical western Pacific Ocean. It has been clarified in the beginning of Section 4 as: "The warm pool of the tropical western Pacific Ocean (130°W-170°E, 20°S-20°N) is a typical region of oceanic convection precipitating and producing anvil clouds (Wall et al., 2018). In this section, only the OS lifecycles over the oceans in this region are considered for investigating the behaviors of the oceanic convection precipitating and producing anvil clouds."

Figure 6 has been added (as shown below) in the revised manuscript to show the sample number of the tracked lifecycles in tropical western Pacific (130°W-170°E, 20°S-20°N) in 2006.

[Figure]

Figure 6. Sample numbers of tracked OS lifecycles with cold-core-peak BT11 values from 185-255 K in the tropical western Pacific (130°W-170°E, 20°S-20°N) in 2006. The contribution fraction of the OS lifecycles to the precipitation and anvil cloud amount is shown on the right axis.

---

## Referee Report (RR1)

**Review of "Observing convective activities in complex convective organizations and their contributions to precipitation and anvil cloud amounts" by Zhenquan Wang and Jian Yuan.**

Thank you to the authors for their responses to my comments. This revision is greatly improved relative to the initial submission. The text is much clearer, and the additional figures and text were very helpful, especially the revised section 3 and addition of Figure 2.

This paper is primarily one about methodology. The convective tracking algorithm is complex and sophisticated, and while the description of it was difficult to follow in the original submission, it is much clearer now, and the reader can understand the methodology being described. I appreciate the large amount of technical work that was done here, and think that this tracking system is potentially very useful to studies of tropical convection.

After describing the tracking algorithm, the authors use it to examine some properties of tracked convective segments in the West Pacific. While I agree that it is a good idea to show some results like this, I found this section to be much less compelling than the methodology itself, as the results are presented rather plainly without much context or discussion. In addition, this section still suffers from lack of clarity in some places, which makes it hard sometimes to understand the results. More details provided below.

**General Comments:**

1. **Clarity of writing.** While the writing in the paper is now clear enough to convey most points, it still does not read easily in many places, and the authors may wish to pursue more editing services if that is an option for them.

2. **Unclear definition of the OS life cycle.** Section 4 presents statistics about OS properties integrated over the OS life cycle. But since an OS is not the same as a whole convective system, it is not clear when the life cycle begins and ends. The lack of detail here prevented me from understanding and contextualizing the results. When exactly does the life cycle begin and end? How is it calculated when there are mergers and splits? For example, for OS #4 in the third row of Fig 5—when does the life cycle start and end in this case? The current explanation in Table 1 (last row) is not particularly helpful for the reader. A schematic may be helpful here.

   (Line 404 / Fig 7a) It is surprising, almost hard to believe, that so many OSs with cold-core-peak BT < 220 K would have a life cycle of just 1 hour. Convection with this BT is near the cold point, and I would expect the life cycle to far exceed 1 hour in almost all cases. Since the authors are using hourly BT images, a 1-hr lifecycle would mean that the OS appears in only a single image, which seems very strange for a convective plume with a BT as cold as 190 K. Is this real, or is this caused by something else (e.g., storms moving out of the study region, or a brief split in the OS before it merges back together). And for a 1-hr lifecycle consisting of just one BT image, how are the development and decay stages defined?

(Line 453 / Fig 9f ) Because the life cycle is not clearly defined, the *N* parameter used in section 4 was confusing. Until *N* was introduced, I thought the results in section 4 were for individual OSs. I was then confused as to how a single OS could contain more than one OS. This must be because two OSs that merge together at a later time are actual considered as the same OS (as in Fig 5). Even so, I am not sure how *N* would be defined in many cases. E.g., if two cores merge together and later split back up, is *N* equal to 1, 2, 3, or 5?

3. **Motivation for section 4.** The results in section 4, if I am understanding correctly, are for individual OSs. It would be helpful to motivate this analysis more clearly. Why do we care about properties of individual OSs, as opposed to properties of entire convective systems?

**Specific Comments:**

- There is no comment or justification about the choice of the 50% threshold for the Dynamic Overlap Ratios (lines 237-245). Is this an arbitrary choice or based on previous work? Do the authors expect the results to be sensitive to this choice? Have there been any sensitivity tests conducted, or at least a tally of how often an OS match fails due to DOR below 50%?

- I understand why the situations in Fig 1c(iii) and Fid 1d(i, iii, and iv) are excluded based on DOR below 50%. These examples are obviously idealized cartoons used to explain the methodology, and I find them quite effective in doing so, but I am curious if situations like these would even be possible in the first place. Consider 1d(iii) as an example, which is the simplest because the structure of the OS is identical between the two time steps. In the first part of the tracking algorithm, cross-correlation is used to predict the location of the OS one hour into the future. This should predict a position that has maximum correlation with the actual BT11 observations from the next time step. So, why would the cross-correlation ever place the OS prediction in the spot that is shown in the figure? Shouldn't the maximum correlation occur when the two solid and dashed lines exactly overlap?

Nothing has to be changes here, since these are just cartoon examples and the real observations are more complex. But I'm curious if these is just an idealized example that would not actually occur, or if I am misunderstanding something about the tracking procedure.

- Line 315-327 / Fig 4a: I've read this paragraph several times and think I mostly understand the point the authors are making here, but I think some of it is still not getting though. I think the authors are just saying that fixed BT thresholds do not capture the structural complexity that still exists in the region where BT is colder than the threshold. But I am not sure how this related to Fig 4a, which I am struggling to understand. Why would most of OSs have cold-center BTs equal to their cold-core BT? Shouldn't the cold-center BTs always be warmer? And if they are indeed equal, wouldn't that mean that most cores are only ~1K colder than the rest of the convective complex? I am pretty sure I am misunderstanding something here, so it would be helpful to clarify this section.

- Line 343-344: "The results in Figs. 4c-e might imply that the OSs of colder cores have increased precipitation efficiency, which contributes to both more precipitation and anvil clouds." I do not

see how the authors can claim that greater precip efficiency leads to greater anvil cloud area. What would be the proposed mechanism for this? Lindzen et al (2001) suggested the exact opposite, although I am not presently aware of any evidence for their claim that does not rely on model microphysics parameterizations. The authors find that storms with lower BT have greater precip efficiency, greater precip area, and greater anvil area. But this might simply mean that storms with lower BT are larger storms. To assess the relationship between precip efficiency and anvil area, one would have to control for BT. I suggest revision of this sentence. Another conclusion could be that the observed relationship between BT and precip efficiency might be expected – storms with higher precip efficiency generally have less dry air entrainment, which may allow updrafts to reach higher altitudes and lower BTs.

 - How exactly is the lifetime-accumulated anvil fraction defined? Are you simply adding up the anvil areas from each BT image? The units are km^2, but if you are measuring area over a period of time, shouldn't the units be hours*km^2?

 - Fig 9 / line 446-450. I imagine that much of the differences in anvil area and precip between four these subgroups can be explained simply by the differences in life cycle duration shown in Fig 9c. The fractional changes in anvil/precip seem to roughly line up with the fractional changes in duration. I would not expect this to be exact of course, but maybe this could explain most of the difference.

 - Line 452: Is the difference in life cycle duration not another mechanism that could explain these differences?

 - Line 467: if A and P are hourly anvil and precip, and N is the total accumulated number of OSs, I do not understand how AN and PN are the lifecycle accumulated A and P. Doesn't there need to be a life cycle duration term in here to achieve that result? E.g., PND, where P is mean hourly precip for a single OS, N is the number of OS, and D is the life cycle duration of each OS?
        I do not doubt that the author's analysis and units are correct, but I think there is a miscommunication or mislabeling here.

 - It would help contextualize the results in section 4 if the frequency of the four life cycle categories are provided somewhere. The authors state that simple life cycle events are rare, but I don't believe the numbers are not actually provided.

 - In section 4, it would be appropriate to remind the readers that "anvil" as defined here still requires BT<260 K. In reality, much the area of detrained cirrus has BT warmer than 260. Berry & Mace 2014 and Sokol & Hartmann 2020 show that anvils with optical depth of 1-2 are extremely common, and Gasparini et al 2022 (DOI: 10.1175/JCLI-D-21-0211.1) showed that these anvils can have BTs warmer than 260.

- An interesting validation experiment for the tracking algorithm could be done using a cloud-resolving model with high-frequency output and a BT11 simulator. "Observations" could be taken from the simulation at every hour, and the tracking could be applied to those

"observations". The tracking results could then be compared to the higher-frequency model output to see if the segments are correctly tracked. This is a big undertaking and is *not* a suggestion for the current paper, simply an idea if the authors ever wished to further validate the algorithm while avoiding the uncertainties associated with the wind observations.

**Minor/Line Comments:**

- Line 311-314: I suggest revising this section, as I am not sure what it is saying after reading it a few times: "The complexity of convective organizations can be inferred from the cold-center BT11 of OSs. Only when the cold-center BT11 is 260 K is the OS of the isolated convective body. Under the fixed BT11 threshold, the OS of the cold-core BT11 that is warmer than the selected threshold cannot be identified. The OS of the cold-center BT11 that is colder than the selected threshold cannot be isolated from CCOs."

- Fig 1c,d: the terminology in the labels is a bit confusing here, and it took me a while to figure out what was being shown. One possible revision is to label the solid lines as "OS position predicted by cross correlation" and the dashed lines as "observed OS position".

 - Line 247-248: "The variation in the cold-core BT11 is prior to the variation…and decay." The use of the word "prior" here was a bit confusing – perhaps "considered first" instead?

 - Line 329: "the colder OS" -> "a colder OS"

- Line 330: "the warmer OS" -> "a warmer OS"

- Line 434-435: "is more distributed". Is there a word missing here between "more" and "distributed"?- Line 480: "anvil production is enhanced" – is the evidence for this just that the $\overline{N}A'$ term is positive? I am just trying to understand.

---

## Author Response (AR2)

**Response to Anonymous Referee #1**

*Referee #1: Thank you to the authors for their responses to my comments. This revision is greatly improved relative to the initial submission. The text is much clearer, and the additional figures and text were very helpful, especially the revised section 3 and addition of Figure 2.*

*This paper is primarily one about methodology. The convective tracking algorithm is complex and sophisticated, and while the description of it was difficult to follow in the original submission, it is much clearer now, and the reader can understand the methodology being described. I appreciate the large amount of technical work that was done here, and think that this tracking system is potentially very useful to studies of tropical convection.*

*After describing the tracking algorithm, the authors use it to examine some properties of tracked convective segments in the West Pacific. While I agree that it is a good idea to show some results like this, I found this section to be much less compelling than the methodology itself, as the results are presented rather plainly without much context or discussion. In addition, this section still suffers from lack of clarity in some places, which makes it hard sometimes to understand the results. More details provided below.*

**Response:** Thank you for reviewing this paper again. We are sincerely grateful for your insightful comments that help us a lot to revise and improve this paper further. We have revised the manuscript carefully according to the comments. The sect. 4 has been better constructed with more discussion and its motivation has been clarified.

*General Comments:*

1. **Clarity of writing.** *While the writing in the paper is now clear enough to convey most points, it still does not read easily in many places, and the authors may wish to pursue more editing services if that is an option for them.*

   **Response:** Thanks. More descriptions have been added in the revised manuscript and the language use and grammar are double checked by editing services.

2. **Unclear definition of the OS life cycle.** *Section 4 presents statistics about OS properties integrated over the OS life cycle. But since an OS is not the same as a whole convective system, it is not clear when the life cycle begins and ends. The lack of detail here prevented me from understanding and contextualizing the results. When exactly does the life cycle begin and end? How is it calculated when there are mergers and splits? For example, for OS #4 in the third row of Fig 5—when does the life cycle start and end in this case? The current explanation in Table 1 (last row) is not particularly helpful for the reader. A schematic may be helpful here.*

   **Response:** A schematic has been added as Fig. 6 in the revised manuscript to help understand the OS and CCO tracking (as illustrated below).

Fig. 6 illustrates an idealized tracking for a CCO and its OSs. The real-world CCO tracking can be much longer and more complicated than that in Fig. 6, and here, it is just used to illustrate how to understand the CCO and OS tracking. As illustrated in Fig. 6, the CCO tracking (dashed black line) can capture the variation in precipitation and anvil areas contributed by multiple convections, but it does not link these variations to specific convections. Mergers and splits in the CCO life cycle reflect the connections and disconnections between different convections. With the OS tracking, the CCO life cycle can be decomposed into the life cycles of its structural components (the colored lines in Fig. 6). It can be recognized that the life cycle of the CCO starts with three convective activities, and with time two of them are merged into the OS1 life cycle and the left one splits into two as the OS2 life cycle. In this way, precipitation and anvil clouds are associated with convective activities in CCOs. On the other hand, the CCO is a large envelope of many convective activities and it is not expected that they all have simple perfect life cycles from convective initiation to anvil dissipation. The OS might just be born from the split of the anvil or the secondary convective activity in its parent stronger convective body (e.g., the OS4 life cycle in Fig. 6) and ends by merging into the anvil in the CCO (e.g., the OS3 life cycle in Fig. 6). The OS tracking documents the life cycle of the core structure from initiation to dissipation. It can be expected that the active convective activities have robust and durable core structures in CCOs, while weak secondary convective activities are fragile and short-lived. In Fig. 7 and Fig. 8, the basic features of OS life cycles of different peaking strengths are investigated for their occurrence, duration and contributions to precipitation and anvil clouds.

The OS lifetime-accumulated precipitation and anvil amount are the sum of the observed OS precipitation and anvil in hourly satellite images during its lifetime. For example, for the OS1 life cycle in Fig. 6, the lifetime-accumulated precipitation and anvil amount are the sum of hourly precipitation and anvil in the OS1 life tree.

This clarification has been added to the revised manuscript.

[Figure]

A CCO life cycle consisting of OS1, OS2, OS3 and OS4

Figure 6. Illustrations of the difference between tracking a CCO and tracking OSs. The CCO life cycle consist of four OSs. The dash black line indicates the tree of the CCO life cycle. The blue, green, red and yellow lines indicate the tree of OS1, OS2, OS3 and OS4 life cycles, respectively.

*(Line 404 / Fig 7a) It is surprising, almost hard to believe, that so many OSs with cold- core-peak BT < 220 K would have a life cycle of just 1 hour. Convection with this BT is near the cold point, and I would expect the life cycle to far exceed 1 hour in almost all cases.*

**Response:** It is not expected that the OS would have a perfect life cycle from convective initiation to anvil dissipation. Some of OSs, even for those very cold structures, are just very short-lived overshooting with only 1-hour duration for the secondary convection in its parent stronger convective body and then disappear or could be annexed by its neighborhood stronger vertical-developing convection in CCOs.

*Since the authors are using hourly BT images, a 1-hr lifecycle would mean that the OS appears in only a single image, which seems very strange for a convective plume with a BT as cold as 190 K. Is this real, or is this caused by something else (e.g., storms moving out of the study region, or a brief split in the OS before it merges back together). And for a 1-hr lifecycle consisting of just one BT image, how are the development and decay stages defined?*

**Response:** The life cycles touching the missing images and edges are excluded from the analyses for quality control. The peaking time is counted as the development stage. Thus, for those short-lived OSs with only 1-hour duration, their development time is just 1 hour and the decay time is zeros. This has been clarified in the revised manuscript.

*(Line 453 / Fig 9f) Because the life cycle is not clearly defined, the N parameter used in section 4 was confusing. Until N was introduced, I thought the results in section 4 were for individual OSs. I was then confused as to how a single OS could contain more than one OS. This must be because two OSs that merge together at a later time are actual considered as the same OS (as in Fig 5). Even so, I am not sure how N would be defined in many cases. E.g., if two cores merge together and later split back up, is N equal to 1, 2, 3, or 5?*

**Response:** The N has been replaced with the lifetime (L) for understanding the impacts of mergers and splits on the precipitation and anvil production. The life cycle with mergers and splits has been illustrated in Fig. 6 in the revised manuscript. Yes, if two cores merge together and later split back up, N is equal to 5. We have revised this part to understand the mergers and splits from the lifetime instead of N. In the OS life cycle, the variation in the accumulated precipitation and anvil cloud amounts can be attributed to two possible factors: (1) the hourly precipitation and anvil production in the life cycle are enhanced by mergers and splits, and (2) the lifetime is prolonged by mergers and splits.

3. ***Motivation for section 4.*** *The results in section 4, if I am understanding correctly, are for individual OSs. It would be helpful to motivate this analysis more clearly. Why do we care about properties of individual OSs, as opposed to properties of entire convective systems?*
**Response:** The motivation for section 4 has been clarified at its beginning as follows: "The total precipitation and anvil cloud amounts of convection are important for tropical water and radiative budgets. They can be attributed to two factors: (1) the occurrence frequency of convection and (2) the precipitation and anvil production for the duration of convection. However, over the warm pool of tropical oceans, convective activities are clustered in CCOs and their precipitation and produced anvil clouds are merged (as discussed in Sect. 3). As a result, identifying their contributions to precipitation and anvil clouds is difficult. On the other hand, the CCO is a large cluster for a series of alternating successive convective activities, which are initiated at different times and evolve in different ways. Thus, there is a dilemma in tracking convection: convection is not isolated naturally for tracking, whereas the CCO is the envelope of many convections whose precipitating and anvil areas are mixed, and it is difficult to identify single convective processes from the CCO life cycle.

It has long been well observed by various active and passive sensors that tropical convections have core structures, e.g., convective pillars observed by active sensors (Igel et al., 2014; Takahashi and Luo, 2012; Deng et al., 2016), heavy raining cores observed by radar or passive microwave radiometer (Yuan and Houze, 2010; Feng et al., 2011), and cold cores of BT11 observed by GEO or MODIS radiometers (Yuan and Houze, 2010; Yang

et al., 2023; La and Messager, 2021). Although convective structures can be better identified by active sensors than by passive sensors, active sensors are only available at a limited number of ground-based sites or on polar-orbit satellites, and their samplings are too sparse for tracking. Yuan and Houze (2010) and Yang et al. (2023) both used active and passive sensors in combination and demonstrated that the BT11 structures are strongly associated with the convective structures. Yuan and Houze (2010) reported that cold and warm BT11 correspond to two distinct types of clouds detected by active sensors: very deep convective clouds and elevated anvil clouds. They partitioned the CCO into single-core high cloud systems (i.e., the OS defined in this work) and identified those OSs with heavy precipitation and the cold-core BT11 colder than 220 K as mesoscale convective systems (MCSs). For these MCSs, Yuan et al. (2011) observed that the cloud vertical structures are well organized, in which high-topped clouds extend outward from raining cores and the thickness of the anvil and the sizes of ice particles are closely related to the distance to the raining cores. Similarly, Yang et al. (2023) identified the cold cores of BT11 as convective centers and also found that the cold-core structures of BT11 are highly consistent with the convective structures detected by active sensors. These findings suggest that cold cores of BT11 can be used to identify the most convective-developing centers and distinguish convective activities in CCOs. In Sect. 3, a novel algorithm was developed to accomplish tracking for convective cold-core structures on the basis of previous studies (Yuan and Houze, 2010; Yuan et al., 2011) and GEO observations.

These OSs can be used to infer different convective activities clustered in CCOs. They are organized differently with various depths of development, precipitation and anvil production and have distinct evolution processes. In this section, on the basis of variable-BT11 tracking, the relationships of convective contributions to precipitation and anvil clouds with their BT11 structural evolution are explored. This would provide an opportunity to compare convections of different development strengths and evolutions for their contributions to precipitation and anvil clouds.".

***Specific Comments:***
*- There is no comment or justification about the choice of the 50% threshold for the Dynamic Overlap Ratios (lines 237-245). Is this an arbitrary choice or based on previous work? Do the authors expect the results to be sensitive to this choice? Have there been any sensitivity tests conducted, or at least a tally of how often an OS match fails due to DOR below 50%?*

**Response:** The 50% threshold for the dynamic overlap ratios is based on the previous work (Williams and Houze, 1987). There is no doubt that the lower the area threshold is, the more easily the OS associations are.

In Fig. 1c-d (as shown below), the occurrence frequency of those overlapping conditions is listed at the top of each subpanel (the blue numbers), in which the red numbers in the parentheses refer to the frequency for conventional stationary overlaps without consideration of the OS movement. Here, every pair of OSs with overlaps is counted as one sample. For instance, if one OS has overlaps with five OSs of the next moment, there would be five pairs of overlapped OSs and the sample number is five. The frequency is

defined as the occurrence of each overlapping condition in Fig. 1c-d divided by the total sample number. It is not surprising that the condition of (ⅲ) in Fig. 1d accounts for the largest portion of samples, since the OSs in CCOs are close to each other and their margins are easily overlapped. As compared with the stationary overlaps, the dynamic overlaps increase the frequency of the overlapping condition of (ⅱ) of Fig. 1c twofold and decrease the frequency of all other conditions. Overall, the dynamic overlaps increase the frequency of associations (the sum of frequency of (ⅰ), (ⅱ) and (ⅳ) in Fig. 1c and (ⅱ) in Fig. 1d) by 2.5% from 21.2% (the frequency of associations for stationary overlaps) to 23.7%.

This has been clarified in the revised manuscript.

[Figure]

Fig. 1c-d. The dynamic overlapping situations of two OSs of different moments when their cores have overlaps and no overlaps, respectively. The solid blue and green lines indicate the core and OS of the current moment at the position predicted by cross correlation. The dashed blue and green lines indicate the core and OS positions of the next moment. The gray cross indicates the non-association between OSs. The occurrence frequency of each condition is listed at the top of each subpanel by blue numbers. The frequency for overlaps without consideration of the OS movement is listed in parentheses by red numbers.

*- I understand why the situations in Fig 1c (iii) and Fid 1d (i, iii, and iv) are excluded based on DOR below 50%. These examples are obviously idealized cartoons used to explain the methodology, and I find them quite effective in doing so, but I am curious if situations like these would even be possible in the first place. Consider 1d (iii) as an example, which is the simplest because the structure of the OS is identical between the two time steps. In the first part of the tracking algorithm, cross-correlation is used to predict the location of the OS one hour into the future. This should predict a position that has maximum correlation with the actual BT11 observations from the next time step. So, why would the cross-correlation ever place the OS prediction in the spot that is shown in the figure? Shouldn't the maximum*

*correlation occur when the two solid and dashed lines exactly overlap?*

*Nothing has to be changes here, since these are just cartoon examples and the real observations are more complex. But I'm curious if these is just an idealized example that would not actually occur, or if I am misunderstanding something about the tracking procedure.*

**Response:**

[Figure]

An OS might have overlaps with many OSs of the next moment simultaneously and the overlapping situations are various. For instance, as shown above, one OS can have large overlaps with the major core structure of an OS of the next moment (such as the overlap with OS1). And meanwhile it also can have some overlaps with the margins of another OS of the next moment (such as the overlap with OS2). The overlap in margins is the condition of (ⅲ) in Fig. 1d. It can happen very frequently since the OSs in CCOs are close to each other and their margins are easily overlapped. The occurrence frequency of each overlapping condition in Fig. 1c-d has been listed in each subpanel (please see our response to the last comment).

More descriptions have been added to clarify the tracking procedures of Fig. 1c-d as follows: "Two OSs of different moments are associated in time and considered the same one OS at different times when these two OSs overlap sufficiently. With the dynamic overlap, an OS is moved to the new predicted location via cross correlation to overlap with the OSs at the next moment. In this case, a necessary condition to consider the associations of the OSs at the next moment to the OS is their DORs at least greater than zeros. After moving it to the new predicted location, an OS might have overlaps with many OSs of the next moment simultaneously and the overlapping situations are various. For instance, one OS can have large overlaps with the major core structure of an OS of the next moment and meanwhile it also can have some overlaps with the margins of another OS of the next moment. Those three DOR indices can be used to identify these distinct overlapping conditions from the overlapping degrees of their cores and OSs, as illustrated in Fig. 1c-d.

The overlapping situations of two OSs are distinguished by whether they have core overlaps (Fig. 1c) or not (Fig. 1d). A sufficient overlapping degree is discriminated by more than 50% of DORs, which is consistent with that in Williams and Houze (1987). If their cores have overlaps, with the DOR between either cores or OSs greater than 50%, the major parts of those pairs of OSs in situations (ⅰ), (ⅱ) and (ⅳ) in Fig. 1b are all sufficiently overlapped,

and thus are associated as the same one OS of different times. The situation (ⅲ) in Fig. 1c with DORs of both cores and OSs less than 50% indicates that these two OSs are only overlapped in margins, without associations in time. In Fig. 1d, when the cores of two OSs are not overlapped, the determinant of the OS association relies on the DOR between OSs and the DOR of OSs to cores. In those cases, the OSs are associated in time only in situation (ⅱ) in Fig. 1d, with large overlaps of their major parts and those two DOR indices both larger than 50%. Those pairs of OSs in the other situations in Fig. 1d are obviously not associated. Overall, if the DORs of two OSs satisfy the overlapping conditions of (ⅰ), (ⅱ) and (ⅳ) in Fig. 1c and (ⅱ) in Fig. 1d, they are associated in time and regarded as the same OS evolving with time.

In Fig. 1c-d, the occurrence frequency of those overlapping conditions is listed at the top of each subpanel (the blue numbers), in which the red numbers in the parentheses refer to the frequency for conventional stationary overlaps without consideration of the OS movement. Here, every pair of OSs with overlaps is counted as one sample. For instance, if one OS has overlaps with five OSs of the next moment, there would be five pairs of overlapped OSs and the sample number is five. The frequency is defined as the occurrence of each overlapping condition in Fig. 1c-d divided by the total sample number. It is not surprising that the condition of (ⅲ) in Fig. 1d accounts for the largest portion of samples, since the OSs in CCOs are close to each other and their margins are easily overlapped. As compared with the stationary overlaps, the dynamic overlaps increase the frequency of the overlapping condition of (ⅱ) of Fig. 1c twofold and decrease the frequency of all other conditions. Overall, the dynamic overlaps increase the frequency of associations (the sum of frequency of (ⅰ), (ⅱ) and (ⅳ) in Fig. 1c and (ⅱ) in Fig. 1d) by 2.5% from 21.2% (the frequency of associations for stationary overlaps) to 23.7%.".

*- Line 315-327 / Fig 4a: I've read this paragraph several times and think I mostly understand the point the authors are making here, but I think some of it is still not getting though. I think the authors are just saying that fixed BT thresholds do not capture the structural complexity that still exists in the region where BT is colder than the threshold. But I am not sure how this related to Fig 4a, which I am struggling to understand. Why would most of OSs have cold-center BTs equal to their cold-core BT? Shouldn't the cold-center BTs always be warmer? And if they are indeed equal, wouldn't that mean that most cores are only ~1K colder than the rest of the convective complex? I am pretty sure I am misunderstanding something here, so it would be helpful to clarify this section.*

**Response:** The cold core and cold center are identified by a set of adaptive thresholds of 180-260 K per 5-K interval. Thus, the cold-core and cold-center BT11 is 180, 185, 190, … 260K. If the cold-core BT11 is 190K, the cold-center BT11 could be 190, 195, … 260K. If the cold-center BT11 is 200K, it means that this OS can be isolated within the 200-K isotherm and there is no need of segmentation (the fixed-threshold identification under 200 K can be used), but in the 205-K or warmer isotherm it would be connected with other OSs (the variable-BT11 identification is needed for segmentation).

There is no doubt that the warmer the selected BT11 threshold is, the more complex the identified target is in the fixed-threshold identification. But, can we just use one cold

BT11 threshold to avoid the complicated connected convective organizations? If it works, the fixed-BT11 tracking under the cold threshold would perform well. The aim of Fig. 4a is to answer this question. For instance, Feng et al. (2018) tried to use two thresholds to identify convective systems with a cold threshold of 225K to capture the cold core and a warm threshold of 241K to find the cloud pixels associated with the cold cores. In this case, is the 225-K cutoff the simple or complicated structure? Fig. 4a gives the answer as shown below.

[Figure]

(1) **For the OSs of the cold-core BT11 from 230-260K**, they would be ignored since these OSs develop warmer than 225K;

(2) **For the OSs of the cold-core BT11 from 190-220K and the cold-center BT11 190-220K**, these OSs would be in complicated convective organizations, which cannot be simply identified by the fixed threshold of 225K;

(3) **For the OSs of the cold-core BT11 from 190-220K and the cold-center BT11 225-260K**, they can be directly isolated by the fixed threshold of 225K, but it only accounts for a small portion of the OSs with cold-core BT11 from 190-220K.

This paragraph has been modified as: "In Fig. 4, the OS structural characteristics (i.e., the connecting conditions with other surrounding OSs in CCOs and their contributions to precipitation and anvil cloud areas) of different development depths with the cold-core BT11 from 190-250 K are investigated. In Fig. 4a, for the OSs of the cold core from 190-250 K, the probability distribution functions (PDFs) of the cold-center BT11 are shown. The cold-core and cold-center BT11 are both identified by 5-K-interval adaptive thresholds (see details in Section 3.1). The PDFs in Fig. 4a have a maximum peak of approximately 36-41% when the cold-center BT11 is equal to the cold-core BT11. This implies that for most of them only the cold core can be isolated by the fixed threshold. For the deep convection of the cold-core

BT11 at 190-220 K, the isolated structure with a cold-center BT11 of 260 K is rare, but it is relatively more frequent and seems to be another mode for the shallow warm systems of the cold-core BT11 at 230-260 K. However, fixed-threshold tracking is not capable of discriminating between isolated and complicated structures.

There is no doubt that the warmer the selected BT11 threshold is, the more complex the identified target is in the fixed-threshold identification. However, can one cold BT11 threshold be used to avoid complicated connected convective organizations? If feasible, the fixed-BT11 tracking under the cold threshold performs well. For instance, Feng et al. (2018) tried to use two thresholds to identify convective systems with a cold threshold of 225 K to capture the cold core and a warm threshold of 241 K to find the cloud pixels associated with the cold cores. In this case, is the 225-K cutoff a simple or complicated structure? If under the fixed threshold of 225 K, Fig. 4a shows that:

1) For the OSs of the cold-core BT11 from 230-260 K, they would be ignored since these OSs develop warmer than 225 K;

2) For the OSs of the cold-core BT11 from 190-220 K and the cold-center BT11 from 190-220 K, they would be in complicated convective organizations, and cannot be simply identified by the fixed threshold of 225 K;

3) For the OSs of the cold-core BT11 from 190-220 K and the cold-center BT11 from 225-260 K, they can be directly isolated by the fixed threshold of 225 K, but it accounts for only a small portion of the OSs of the cold-core BT11 from 190-220 K.

This implies that even under the cold BT11 threshold, most of the identified targets still have complex organizations.".

*- Line 343-344: "The results in Figs. 4c-e might imply that the OSs of colder cores have increased precipitation efficiency, which contributes to both more precipitation and anvil clouds." I do not see how the authors can claim that greater precip efficiency leads to greater anvil cloud area. What would be the proposed mechanism for this? Lindzen et al (2001) suggested the exact opposite, although I am not presently aware of any evidence for their claim that does not rely on model microphysics parameterizations. The authors find that storms with lower BT have greater precip efficiency, greater precip area, and greater anvil area. But this might simply mean that storms with lower BT are larger storms. To assess the relationship between precip efficiency and anvil area, one would have to control for BT. I suggest revision of this sentence. Another conclusion could be that the observed relationship between BT and precip efficiency might be expected – storms with higher precip efficiency generally have less dry air entrainment, which may allow updrafts to reach higher altitudes and lower BTs.*

**Response:** Thanks. The previous statement has been removed. A new conclusion according to the reviewer's comment has been added: "the observed relationship between BT11 structures and precipitation efficiency might be expected. Storms with higher precipitation efficiency generally have less dry air entrainment, which may allow updrafts to reach higher altitudes and lower BT11.".

*- How exactly is the lifetime-accumulated anvil fraction defined? Are you simply adding up the anvil areas from each BT image? The units are km^2, but if you are measuring area over*

*a period of time, shouldn't the units be hours\*km^2?*

**Response:** Yes, the lifetime-accumulated anvil areas are computed by adding up the anvil areas from each hourly BT image during the lifetime. The unit of the anvil area in each hourly BT images is km^2 and thus the sum of it over a period of time is km^2\*hour. The units have been corrected in the revised manuscript.

*- Fig 9 / line 446-450. I imagine that much of the differences in anvil area and precip between four these subgroups can be explained simply by the differences in life cycle duration shown in Fig 9c. The fractional changes in anvil/precip seem to roughly line up with the fractional changes in duration. I would not expect this to be exact of course, but maybe this could explain most of the difference.*

**Response:** Yes, the changes in the duration are very important for explaining the differences in anvil area and precipitation.

This paragraph has been modified as: "How do mergers and splits influence the lifecycle-accumulated precipitation and anvil cloud amounts? This question is simply explored from the OS tracking. In the OS life cycle, the variation in the accumulated precipitation and anvil cloud amounts can be attributed to two possible factors: (1) the hourly precipitation and anvil production in the life cycle are enhanced by mergers and splits, and (2) the lifetime is prolonged by mergers and splits.

In Figs. 11a-b, the hourly mean precipitation and anvil amount in the OS life cycles are shown for different types of life cycles. For the same cold-core-peak BT11, the hourly mean precipitation of different lifecycle types is nearly invariant (Fig. 11a). However, in the life cycles with the occurrence of mergers and splits, the hourly mean anvil production is enhanced (Fig. 11b), and the lifetime (L) is significantly prolonged (Fig. 10c). To quantify their impacts, in Figs. 11c-d, the anomalies of the lifetime-accumulated precipitation and anvil cloud amounts can be decomposed as follows:

$$PL - \bar{P}\bar{L} = \bar{L}P' + \bar{P}L' + P'L', \tag{7}$$
$$AL - \bar{A}\bar{L} = \bar{L}A' + \bar{A}L' + A'L'. \tag{8}$$

P and A are the hourly precipitation and anvil cloud amount, respectively. L is the lifetime. Thus, PL and AL represent the lifetime-accumulated precipitation and anvil cloud amount, respectively. The bar over the letter represents the mean of different lifecycle types. The prime over the letter represents the anomaly of different lifecycle types relative to their mean value. In this way, $\bar{L}P'$ and $\bar{P}L'$ indicate the contributions of the hourly precipitation anomaly and the lifetime anomaly, respectively, to the variation in lifetime-accumulated precipitation. Similarly, $\bar{L}A'$ and $\bar{A}L'$ indicate the contributions of the hourly anvil production and lifetime anomalies, respectively, to the variation in the lifetime-accumulated anvil amount. $P'L'$ and $A'L'$ are high-order small quantities and are neglected. The fraction of the contribution can be computed by dividing the left-hand-side quantities of Eq. 7 and Eq. 8. Fig. 11c (Fig. 11d) shows the fractions of the contributions of $\bar{L}P'$ and $\bar{P}L'$ ($\bar{L}A'$ and $\bar{A}L'$) to the increase in lifetime-accumulated precipitation (anvil) from simple to complicated life cycles. For the life cycles of the cold-core-peak BT11 colder than 220 K, $\bar{L}P'$ has a relatively small contribution of approximately 10-25%, whereas $\bar{P}L'$ has a large contribution of approximately 60-80%. In

addition, $\bar{L}A'$ and $\bar{A}L'$ both have positive comparable contribution fractions, approximately 20-40% and 40-60%, respectively. For the warmer life cycles, the contributions from $\bar{L}P'$ and $\bar{L}A'$ increase and are more important than the lifetime anomaly for the variation in the lifetime-accumulated precipitation and anvil cloud amounts.

On average, in comparison with simple life cycles, mergers and splits can significantly prolong the duration of OSs while enhancing the hourly precipitation slightly and increasing the hourly anvil production strongly. From simple to complicated life cycles, a prolonged lifetime accounts for the largest contribution to the increase in accumulated precipitation and anvil clouds for cold structures.".

[Figure]

Figure 11. Composites of the hourly mean precipitation (a) and anvil cloud amounts (b) of different lifecycle types in each bin of the cold-core-peak BT11, respectively. The blue, red, yellow and purple lines indicate the simple, only-merger, only-split and complicated life cycles, respectively. (c) The fractions of contributions of the hourly precipitation anomalies ($\bar{L}P'$) and the lifetime anomalies ($\bar{P}L'$) to the variation in lifetime-accumulated precipitation. (d) The fractions of contributions of the hourly anvil production anomalies ($\bar{L}A'$) and the lifetime anomalies ($\bar{A}L'$) to the variation in the lifetime-accumulated anvil amount. The error bars indicate the 95% confidence intervals of the means based on the t test.

*- Line 452: Is the difference in life cycle duration not another mechanism that could explain these differences?*

**Response:** Yes, the changes in the duration are very important and account for the largest contribution to the variation of the accumulated precipitation and anvil clouds. This paragraph has been revised and please see our responses to the last comment.

*- Line 467: if A and P are hourly anvil and precip, and N is the total accumulated number of OSs, I do not understand how AN and PN are the lifecycle accumulated A and P. Doesn't there need to be a life cycle duration term in here to achieve that result? E.g., PND, where P is mean hourly precip for a single OS, N is the number of OS, and D is the life cycle duration of each OS?*

*I do not doubt that the author's analysis and units are correct, but I think there is a miscommunication or mislabeling here.*

**Response:** Thanks. We have decomposed the lifetime-accumulated precipitation and anvil areas into the hourly mean precipitation and anvil and lifetime, respectively, to explain the variation in the precipitation and anvil clouds in the OS life cycles. Please see our responses to the comments on Fig. 9.

*- It would help contextualize the results in section 4 if the frequency of the four life cycle categories are provided somewhere. The authors state that simple life cycle events are rare, but I don't believe the numbers are not actually provided.*

**Response:** The simple life cycle without mergers and splits is the most frequent and accounts for 93.9% of samples. The only-merger, only-split and complicated life cycle have the frequency of only 3.0%, 1.4% and 1.7%, respectively. This has been provided in the Section 4.3 in the revised manuscript.

*- In section 4, it would be appropriate to remind the readers that "anvil" as defined here still requires BT<260. In reality, much the area of detrained cirrus has BT warmer than 260. Berry & Mace 2014 and Sokol & Hartmann 2020 show that anvils with optical depth of 1-2 are extremely common, and Gasparini et al 2022 (DOI: 10.1175/JCLI-D-21-0211.1) showed that these anvils can have BTs warmer than 260.*

**Response:** This has been added at the beginning of Section 4 to remind the readers about the anvil definition in this work, as follows: "Notably, the anvil identification requires that the BT11 is colder than 260 K and the precipitation is less than 1 mm/hour. It can be used to reflect the anvil productivity in the convective systems (Yuan and Houze, 2010; Yuan et al., 2011; Yuan and Houze, 2013), but much the area of detained cirrus has the BT11 warmer than 260 K in reality (Gasparini et al., 2022; Sokol and Hartmann, 2020; Berry and Mace, 2014). Normally, those thin cirrus clouds are not well identified by GEO radiometers and thus in this work, the anvil just refers to the thick anvil portion identified by the 260-K BT11 threshold but not all detrained anvil cirrus clouds.".

*- An interesting validation experiment for the tracking algorithm could be done using a cloud-resolving model with high-frequency output and a BT11 simulator. "Observations" could be taken from the simulation at every hour, and the tracking could be applied to those*

*"observations". The tracking results could then be compared to the higher-frequency model output to see if the segments are correctly tracked. This is a big undertaking and is not a suggestion for the current paper, simply an idea if the authors ever wished to further validate the algorithm while avoiding the uncertainties associated with the wind observations.*

**Response:** Thanks. We believe it is a very constructive and interesting idea for our future work to further evaluate this tracking algorithm from cloud-resolving models and to apply this algorithm to compare the difference between observed and simulated life cycles. Thanks to the reviewer again for your precious insightful comments.

*Minor/Line Comments:*
*- Line 311-314: I suggest revising this section, as I am not sure what it is saying after reading it a few times: "The complexity of convective organizations can be inferred from the cold-center BT11 of OSs. Only when the cold-center BT11 is 260 K is the OS of the isolated convective body. Under the fixed BT11 threshold, the OS of the cold-core BT11 that is warmer than the selected threshold cannot be identified. The OS of the cold-center BT11 that is colder than the selected threshold cannot be isolated from CCOs.".*

**Response:** This sentence has been deleted and more specific descriptions have been added:

"There is no doubt that the warmer the selected BT11 threshold is, the more complex the identified target is in the fixed-threshold identification. However, can one cold BT11 threshold be used to avoid complicated connected convective organizations? If feasible, the fixed-BT11 tracking under the cold threshold performs well. For instance, Feng et al. (2018) tried to use two thresholds to identify convective systems with a cold threshold of 225 K to capture the cold core and a warm threshold of 241 K to find the cloud pixels associated with the cold cores. In this case, is the 225-K cutoff a simple or complicated structure? If under the fixed threshold of 225 K, Fig. 4a shows that:
1) For the OSs of the cold-core BT11 from 230-260 K, they would be ignored since these OSs develop warmer than 225 K;
2) For the OSs of the cold-core BT11 from 190-220 K and the cold-center BT11 from 190-220 K, they would be in complicated convective organizations, and cannot be simply identified by the fixed threshold of 225 K;
3) For the OSs of the cold-core BT11 from 190-220 K and the cold-center BT11 from 225-260 K, they can be directly isolated by the fixed threshold of 225 K, but it accounts for only a small portion of the OSs of the cold-core BT11 from 190-220 K.
This implies that even under the cold BT11 threshold, most of the identified targets still have complex organizations.".

*- Fig 1c,d: the terminology in the labels is a bit confusing here, and it took me a while to figure out what was being shown. One possible revision is to label the solid lines as "OS position predicted by cross correlation" and the dashed lines as "observed OS position".*

**Response:** Thanks. The labels in Fig. 1c and 1d have been modified as "OS position predicted by cross correlation" and "Observed OS position at the next moment", as shown below.

————— OS core position predicted by cross correlation    – – – Observed OS core position at the next moment

————— OS position predicted by cross correlation    – – – Observed OS position at the next moment

*- Line 247-248: "The variation in the cold-core BT11 is prior to the variation...and decay." The use of the word "prior" here was a bit confusing – perhaps "considered first" instead?*

**Response:** It has been corrected.

*- Line 329: "the colder OS" -> "a colder OS"*

**Response:** It has been corrected.

*- Line 330: "the warmer OS" -> "a warmer OS"*

**Response:** It has been corrected.

*- Line 434-435: "is more distributed". Is there a word missing here between "more" and "distributed"?*

**Response:** It has been revised as: "the cold-core BT11 of mergers is distributed at colder BT11 values than that of splits"

*- Line 480: "anvil production is enhanced" – is the evidence for this just that the $\overline{N}A'$ term is positive? I am just trying to understand.*

**Response:** Yes, the term of $\overline{N}A'$ has a positive contribution. And according to the Fig. 11 in the revised manuscript (or the Fig. 9 in the previous manuscript), for the life cycles from simple to complicated, the hourly anvil production is gradually enhanced with the occurrence of mergers and splits.

**Reference**

Berry, E. and Mace, G. G.: Cloud properties and radiative effects of the Asian summer monsoon derived from A-Train data, Journal of Geophysical Research: Atmospheres, 119, 9492-9508, 10.1002/2014jd021458, 2014.

Deng, M., Mace, G. G., and Wang, Z.: Anvil Productivities of Tropical Deep Convective Clusters and Their Regional Differences, Journal of the Atmospheric Sciences, 73, 3467-3487, 10.1175/jas-d-15-0239.1, 2016.

Feng, Z., Dong, X., Xi, B., Schumacher, C., Minnis, P., and Khaiyer, M.: Top-of-atmosphere radiation budget of convective core/stratiform rain and anvil clouds from deep convective systems, Journal of Geophysical Research: Atmospheres, 116, n/a-n/a, 10.1029/2011jd016451, 2011.

Feng, Z., Leung, L. R., Houze, R. A., Hagos, S., Hardin, J., Yang, Q., Han, B., and Fan, J.: Structure and Evolution of Mesoscale Convective Systems: Sensitivity to Cloud Microphysics in Convection-Permitting Simulations Over the United States, Journal of Advances in Modeling

Earth Systems, 10, 1470-1494, 10.1029/2018ms001305, 2018.

Gasparini, B., Sokol, A. B., Wall, C. J., Hartmann, D. L., and Blossey, P. N.: Diurnal Differences in Tropical Maritime Anvil Cloud Evolution, Journal of Climate, 35, 1655-1677, 10.1175/jcli-d-21-0211.1, 2022.

Igel, M. R., Drager, A. J., and van den Heever, S. C.: A CloudSat cloud object partitioning technique and assessment and integration of deep convective anvil sensitivities to sea surface temperature, Journal of Geophysical Research: Atmospheres, 119, 10515-10535, 10.1002/2014jd021717, 2014.

La, T. V. and Messager, C.: Convective System Observations by LEO and GEO Satellites in Combination, IEEE Journal of Selected Topics in Applied Earth Observations and Remote Sensing, 14, 11814-11823, 10.1109/jstars.2021.3127401, 2021.

Sokol, A. B. and Hartmann, D. L.: Tropical Anvil Clouds: Radiative Driving Toward a Preferred State, Journal of Geophysical Research: Atmospheres, 125, 10.1029/2020jd033107, 2020.

Takahashi, H. and Luo, Z.: Where is the level of neutral buoyancy for deep convection?, Geophysical Research Letters, 39, 10.1029/2012gl052638, 2012.

Williams, M. and Houze, R. A.: Satellite-Observed Characteristics of Winter Monsoon Cloud Clusters, Monthly Weather Review, 115, 505-519, 10.1175/1520-0493(1987)115<0505:Socowm>2.0.Co;2, 1987.

Yang, K., Wang, Z., Deng, M., and Dettmann, B.: Combining CloudSat/CALIPSO and MODIS measurements to reconstruct tropical convective cloud structure, Remote Sensing of Environment, 287, 10.1016/j.rse.2023.113478, 2023.

Yuan, J. and Houze, R. A.: Global Variability of Mesoscale Convective System Anvil Structure from A-Train Satellite Data, Journal of Climate, 23, 5864-5888, 10.1175/2010jcli3671.1, 2010.

Yuan, J. and Houze, R. A.: Deep Convective Systems Observed by A-Train in the Tropical Indo-Pacific Region Affected by the MJO, Journal of the Atmospheric Sciences, 70, 465-486, 10.1175/jas-d-12-057.1, 2013.

Yuan, J., Houze, R. A., and Heymsfield, A. J.: Vertical Structures of Anvil Clouds of Tropical Mesoscale Convective Systems Observed by CloudSat, Journal of the Atmospheric Sciences, 68, 1653-1674, 10.1175/2011jas3687.1, 2011.